# Skill or Luck? Return Decomposition via Advantage Functions

**Hsiao-Ru Pan & Bernhard Schökopf**
Max Planck Institute for Intelligent Systems, Tübingen, Germany

## Abstract

Learning from off-policy data is essential for sample-efficient reinforcement learning. In the present work, we build on the insight that the advantage function can be understood as the causal effect of an action on the return, and show that this allows us to decompose the return of a trajectory into parts caused by the agent's actions (skill) and parts outside of the agent's control (luck). Furthermore, this decomposition enables us to naturally extend Direct Advantage Estimation (DAE) to off-policy settings (Off-policy DAE). The resulting method can learn from off-policy trajectories without relying on importance sampling techniques or truncating off-policy actions. We draw connections between Off-policy DAE and previous methods to demonstrate how it can speed up learning and when the proposed off-policy corrections are important. Finally, we use the MinAtar environments to illustrate how ignoring off-policy corrections can lead to suboptimal policy optimization performance.

## 1 Introduction

Imagine the following scenario: One day, A and B both decide to purchase a lottery ticket, hoping to win the grand prize. Each of them chose their favorite set of numbers, but only A got *lucky* and won the million-dollar prize. In this story, we are likely to say that A got *lucky* because, while A's action (picking a set of numbers) led to the reward, the expected rewards are the same for both A and B (assuming the lottery is fair), and A was ultimately rewarded due to something outside of their control.

This shows that, in a decision-making problem, the return is not always determined solely by the actions of the agent, but also by the randomness of the environment. Therefore, for an agent to correctly distribute credit among its actions, it is crucial that the agent is able to reason about the effect of its actions on the rewards and disentangle it from factors outside its control. This is also known as the problem of credit assignment (Minsky, 1961). While attributing *luck* to the drawing process in the lottery example may be easy, it becomes much more complex in sequential settings, where multiple actions are involved and rewards are delayed.

The key observation of the present work is that we can treat the randomness of the environment as actions from an imaginary agent, whose actions determine the future of the decision-making agent. Combining this with the idea that the advantage function can be understood as the causal effect of an action on the return (Pan et al., 2022), we show that the return can be decomposed into parts caused by the agent (skill) and parts that are outside the agent's control (luck). Furthermore, we show that this decomposition admits a natural way to extend Direct Advantage Estimation (DAE), an on-policy multi-step learning method, to off-policy settings (Off-policy DAE). The resulting method makes minimal assumptions about the behavior policy and shows strong empirical performance.

Our contributions can be summarized as follows:

- We generalize DAE to off-policy settings.
- We demonstrate that (Off-policy) DAE can be seen as generalizations of Monte-Carlo (MC) methods that utilize sample trajectories more efficiently.
- We verify empirically the importance of the proposed off-policy corrections through experiments in both deterministic and stochastic environments.

## 2 BACKGROUND

In this work, we consider a discounted Markov Decision Process $(\mathcal{S}, \mathcal{A}, p, r, \gamma)$ with finite state space $\mathcal{S}$, finite action space $\mathcal{A}$, transition probability $p(s'|s,a)$, expected reward function $r : \mathcal{S} \times \mathcal{A} \to \mathbb{R}$, and discount factor $\gamma \in [0, 1)$. A policy is a function $\pi : \mathcal{S} \to \Delta(\mathcal{A})$ which maps states to distributions over the action space. The goal of reinforcement learning (RL) is to find a policy that maximizes the expected return, $\pi^* = \arg\max_\pi \mathbb{E}_\pi[G]$, where $G = \sum_{t=0}^\infty \gamma^t r_t$ and $r_t = r(s_t, a_t)$. The value function of a state is defined by $V^\pi(s) = \mathbb{E}_\pi[G|s_0{=}s]$, the $Q$-function of a state-action pair is similarly defined by $Q^\pi(s,a) = \mathbb{E}_\pi[G|s_0{=}s, a_0{=}a]$ (Sutton et al., 1998). These functions quantify the expected return of a given state or state-action pair by following a given policy $\pi$, and are useful for policy improvements. They are typically unknown and are learned via interactions with the environment.

**Direct Advantage Estimation**    The advantage function, defined by $A^\pi(s,a) = Q^\pi(s,a) - V^\pi(s)$, is another function that is useful to policy optimization. Recently, Pan et al. (2022) showed that the advantage function can be understood as the causal effect of an action on the return, and is more stable under policy variations (under mild assumptions) compared to the $Q$-function. They argued that it might be an easier target to learn when used with function approximation, and proposed Direct Advantage Estimation (DAE), which estimates the advantage function directly by

$$A^\pi = \underset{\hat{A} \in F_\pi}{\arg\min} \, \mathbb{E}_\pi \left[ \left( \sum_{t=0}^\infty \gamma^t (r_t - \hat{A}_t) \right)^2 \right], \quad F_\pi = \left\{ f \, \middle| \, \sum_{a \in \mathcal{A}} f(s,a)\pi(a|s) = 0 \right\} \quad (1)$$

where $\hat{A}_t = \hat{A}(s_t, a_t)$. The method can also be seamlessly combined with a bootstrapping target to perform multi-step learning by iteratively minimizing the constrained squared error

$$L(\hat{A}, \hat{V}) = \mathbb{E}_\pi \left[ \left( \sum_{t=0}^{n-1} \gamma^t (r_t - \hat{A}_t) + \gamma^n V_{\text{target}}(s_n) - \hat{V}(s_0) \right)^2 \right] \quad \text{subject to } \hat{A} \in F_\pi, \quad (2)$$

where $V_{\text{target}}$ is the bootstrapping target, and $(\hat{V}, \hat{A})$ are estimates of the value function and the advantage function. Policy optimization results were reported to improve upon generalized advantage estimation (Schulman et al., 2015b), a strong baseline for on-policy methods. One major drawback of DAE, however, is that it can only estimate the advantage function for on-policy data (note that the expectation and the constraints share the same policy). This limits the range of applications of DAE to on-policy scenarios, which tend to be less sample efficient.

**Multi-step learning**    In RL, we often update estimates of the value functions based on previous estimates (e.g., TD(0), SARSA (Sutton et al., 1998)). These methods, however, can suffer from excessive bias when the previous estimates differ significantly from the true value functions, and it was shown that such bias can greatly impact the performance when used with function approximators (Schulman et al., 2015b). One remedy is to extend the backup length, that is, instead of using one-step targets such as $r(s_0, a_0) + \gamma Q_{\text{target}}(s_1, a_1)$ ($Q_{\text{target}}$ being our previous estimate), we include more rewards along the trajectory, i.e., $r(s_0, a_0) + \gamma r(s_1, a_1) + \gamma^2 r(s_2, a_2) + ... + \gamma^n Q_{\text{target}}(s_n, a_n)$. This way, we can diminish the impact of $Q_{\text{target}}$ by the discount factor $\gamma^n$. However, using the rewards along the trajectory relies on the assumption that the samples are on-policy (i.e., the behavior policy is the same as the target policy). To extend such methods to off-policy settings often requires techniques such as importance sampling (Munos et al., 2016; Rowland et al., 2020) or truncating (diminishing) off-policy actions (Precup et al., 2000; Watkins, 1989), which can suffer from high variance or low data utilization with long backup lengths. Surprisingly, empirical results have shown that ignoring off-policy corrections can still lead to substantial speed-ups and is widely adapted in modern deep RL algorithms (Hernandez-Garcia & Sutton, 2019; Hessel et al., 2018; Gruslys et al., 2017).

## 3 RETURN DECOMPOSITION

From the lottery example in Section 1, we observe that, stochasticity of the return can come from two sources, namely, (1) the stochastic policy employed by the agent (picking numbers), and (2)

the stochastic transitions of the environment (lottery drawing). To separate their effect, we begin by studying deterministic environments where the only source of stochasticity comes from the agent's policy. Afterward, we demonstrate why DAE fails when transitions are stochastic, and introduce a simple fix which generalizes DAE to off-policy settings.

## 3.1 THE DETERMINISTIC CASE

First, for deterministic environments, we have $s_{t+1} = h(s_t, a_t)$, where the transition probability is replaced by a deterministic transition function $h : \mathcal{S} \times \mathcal{A} \to \mathcal{S}$. As a consequence, the $Q$-function becomes $Q^\pi(s_t, a_t) = r(s_t, a_t) + \gamma V^\pi(s_{t+1})$, and the advantage function becomes $A^\pi(s_t, a_t) = r(s_t, a_t) + \gamma V^\pi(s_{t+1}) - V^\pi(s_t)$. Let's start by examining the sum of the advantage function along a given trajectory $(s_0, a_0, s_1, a_1, ...)$ with return $G$,

$$\sum_{t=0}^{\infty} \gamma^t A^\pi(s_t, a_t) = \sum_{t=0}^{\infty} \gamma^t r(s_t, a_t) + \underbrace{\sum_{t=0}^{\infty} \gamma^t \left( \gamma V^\pi(s_{t+1}) - V^\pi(s_t) \right)}_{\text{telescoping series}} = G - V^\pi(s_0), \quad (3)$$

or, with a simple rearrangement, $G = V^\pi(s_0) + \sum_{t=0}^{\infty} \gamma^t A^\pi(s_t, a_t)$. One intuitive interpretation of this equation is: The return of a trajectory is equal to the average return ($V^\pi$) plus the variations caused by the actions along the trajectory ($A^\pi$). Since Equation 3 holds for *any* trajectory, the following equation holds for *any* policy $\mu$

$$\mathbb{E}_\mu \left[ \left( G - \sum_{t=0}^{\infty} \gamma^t A_t^\pi - V^\pi(s_0) \right)^2 \right] = 0. \quad (4)$$

This means that $(V^\pi, A^\pi)$ is a solution to the off-policy variant of DAE

$$L(\hat{A}, \hat{V}) = \mathbb{E}_\mu \left[ \left( \sum_{t=0}^{\infty} \gamma^t (r_t - \hat{A}_t) - \hat{V}(s_0) \right)^2 \right] \quad \text{s.t.} \sum_{a \in \mathcal{A}} \pi(a|s) \hat{A}(s, a) = 0 \; \forall s \in \mathcal{S}, \quad (5)$$

where the expectation is now taken with respect to an arbitrary behavior policy $\mu$ instead of the target policy $\pi$ in the constraint (Equation 2, with $n \to \infty$). We emphasize that this is a very general result, as we made no assumptions on the behavior policy $\mu$, and only sample trajectories from $\mu$ are required to compute the squared error. However, two questions remain: (1) Is the solution unique? (2) Does this hold for stochastic environments? We shall answer these questions in the next section.

## 3.2 THE STOCHASTIC CASE

The major difficulty in applying the above argument to stochastic environments is that the telescoping sum (Equation 3) no longer holds because $A^\pi(s_t, a_t) = r(s_t, a_t) + \gamma \mathbb{E}_{s' \sim p(\cdot|s_t, a_t)}[V^\pi(s')|s_t, a_t] - V^\pi(s_t) \neq r(s_t, a_t) + \gamma V^\pi(s_{t+1}) - V^\pi(s_t)$ and the sum of the advantage function becomes

$$\sum_{t=0}^{\infty} \gamma^t A^\pi(s_t, a_t) = \sum_{t=0}^{\infty} \gamma^t \left( r(s_t, a_t) + \gamma \mathbb{E}_{s' \sim p(\cdot|s_t, a_t)}[V^\pi(s')|s_t, a_t] - V^\pi(s_t) \right) \quad (6)$$

$$= G - \sum_{t=0}^{\infty} \gamma^{t+1} B_t^\pi - V^\pi(s_0), \quad (7)$$

where $B_t^\pi = B^\pi(s_t, a_t, s_{t+1}) = V^\pi(s_{t+1}) - \mathbb{E}_{s' \sim p(\cdot|s_t, a_t)}[V^\pi(s')|s_t, a_t]$. This shows that $V^\pi$ and $A^\pi$ are not enough to fully characterize the return $G$ (compared to Equation 3), and $B^\pi$ is required. But what exactly is $B^\pi$? To understand the meaning of $B^\pi$, we begin by dissecting state transitions into a two-step process, see Figure 1. In this view, we introduce an imaginary agent *nature*, also interacting with the environment, whose actions determine the next states of the decision-making agent. In this setting, nature follows a stationary policy $\bar{\pi}$ equal to the transition probability, i.e., $\bar{\pi}(s'|(s, a)) = p(s'|s, a)$. Since $\bar{\pi}$ is fixed, we omit it in the following discussion. The question we are interested in is, how much do nature's actions affect the return? We note that, while there

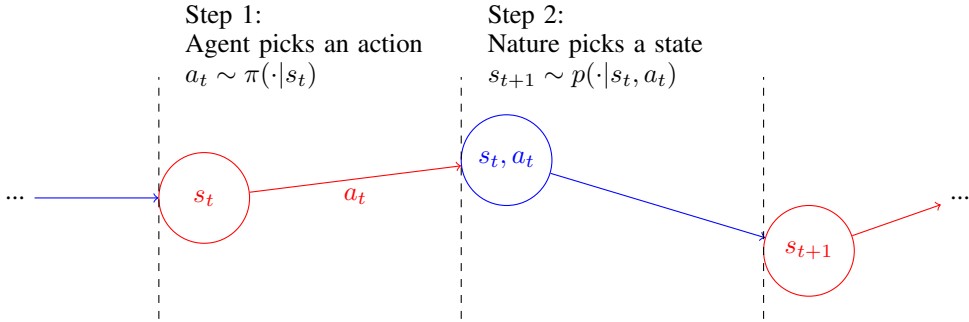

Figure 1: A two-step view of the state transition process. First, we introduce an imaginary agent *nature*, which controls the stochastic part of the transition process. In this view, nature lives in a world with state space $\bar{\mathcal{S}} = \mathcal{S} \times \mathcal{A}$ and action space $\bar{\mathcal{A}} = \mathcal{S}$. At each time step $t$, the agent chooses its action $a_t$ based on $s_t$, and, instead of transitioning directly into the next state, it transitions into an intermediate state denoted $(s_t, a_t) \in \bar{\mathcal{S}}$, where nature chooses the next state $s_{t+1} \in \bar{\mathcal{A}}$ based on $(s_t, a_t)$. We use nodes and arrows to represent states and actions by the agent (red) and nature (blue).

are no immediate rewards associated with nature's actions, they can still influence future rewards by choosing whether we transition into high-rewarding states or otherwise. Since the advantage function was shown to characterize the causal effect of actions on the return, we now examine nature's advantage function.

By definition, the advantage function is equal to $Q^\pi(s, a) - V^\pi(s)$. We first compute both $\bar{Q}^\pi$ and $\bar{V}^\pi$ from nature's point of view (we use the bar notation to differentiate between nature's view and the agent's view). Since $\bar{\mathcal{S}} = \mathcal{S} \times \mathcal{A}$ and $\bar{\mathcal{A}} = \mathcal{S}$, $\bar{V}$ is now a function of $\bar{\mathcal{S}} = \mathcal{S} \times \mathcal{A}$, and $\bar{Q}$ is a function of $\bar{\mathcal{S}} \times \bar{\mathcal{A}} = \mathcal{S} \times \mathcal{A} \times \mathcal{S}$, taking the form

$$\bar{V}^\pi(s, a) = \mathbb{E}_\pi[\sum_{t>0} \gamma^t r_t | s_0{=}s, a_0{=}a] = \mathbb{E}_{s' \sim p(\cdot|s_0, a_0)}[V^\pi(s')|s_0{=}s, a_0{=}a], \tag{8}$$

$$\bar{Q}^\pi(s, a, s') = \mathbb{E}_\pi[\sum_{t>0} \gamma^t r_t | s_0{=}s, a_0{=}a, s_1{=}s'] = V^\pi(s'). \tag{9}$$

We thus have $\bar{A}^\pi(s, a, s') = \bar{Q}^\pi(s, a, s') - \bar{V}^\pi(s, a) = V^\pi(s') - \mathbb{E}_{s' \sim p(\cdot|s,a)}[V^\pi(s')|s, a]$, which is exactly $B^\pi(s, a, s')$ as introduced at the beginning of this section. Now, if we rearrange Equation 6 into

$$V^\pi(s_0) + \sum_{t=0}^{\infty} \gamma^t \left(A^\pi(s_t, a_t) + \gamma B^\pi(s_t, a_t, s_{t+1})\right) = G, \tag{10}$$

then an intuitive interpretation emerges, which reads: *The return of a trajectory can be decomposed into the average return $V^\pi(s_0)$, the causal effect of the agent's actions $A^\pi(s_t, a_t)$ (**skill**), and the causal effect of nature's actions $B^\pi(s_t, a_t, s_{t+1})$ (**luck**).*

Equation 10 has several interesting applications. For example, the policy improvement lemma (Kakade & Langford, 2002), which relates value functions of different policies by $V^\mu(s) = V^\pi(s) + \mathbb{E}_\mu[\sum_{t \geq 0} \gamma^t A_t^\pi | s_0 = s]$, is an immediate consequence of taking the conditional expectation $\mathbb{E}_\mu[\cdot|s_0{=}s]$ of Equation 10. More importantly, this equation admits a natural generalization of DAE to off-policy settings:

**Theorem 1** (Off-policy DAE). *Given a behavior policy $\mu$, a target policy $\pi$, and backup length $n \geq 0$. Let $\hat{A}_t = \hat{A}(s_t, a_t)$, $\hat{B}_t = \hat{B}(s_t, a_t, s_{t+1})$, and the objective function*

$$L(\hat{A}, \hat{B}, \hat{V}) = \mathbb{E}_\mu\left[\left(\sum_{t=0}^{n} \gamma^t \left(r_t - \hat{A}_t - \gamma \hat{B}_t\right) + \gamma^{n+1} \hat{V}(s_{n+1}) - \hat{V}(s_0)\right)^2\right] \tag{11}$$

$$subject\ to \begin{cases} \sum_{a \in \mathcal{A}} \hat{A}(s, a) \pi(a|s) = 0 & \forall s \in \mathcal{S} \\ \sum_{s' \in \mathcal{S}} \hat{B}(s, a, s') p(s'|s, a) = 0 & \forall (s, a) \in \mathcal{S} \times \mathcal{A} \end{cases},$$

*then $(A^\pi, B^\pi, V^\pi)$ is a minimizer of the above problem. Furthermore, the minimizer is unique if $\mu$ is sufficiently explorative (i.e., non-zero probability of reaching all possible transitions $(s, a, s')$).*

See Appendix A for a proof. In practice, we can minimize the empirical variant of Equation 11 from samples to estimate $(V^\pi, A^\pi, B^\pi)$, which renders this an off-policy multi-step method. We highlight two major differences between this method and other off-policy multi-step methods. (1) Minimal assumptions on the behavior policy are made, and no knowledge of the behavior policy is required during training (in contrast to importance sampling methods). (2) It makes use of the full trajectory instead of truncating or diminishing future steps when off-policy actions are encountered (Watkins, 1989; Precup et al., 2000). We note, however, that applying this method in practice can be non-trivial due to the constraint $\sum_{s' \in \mathcal{S}} \hat{B}(s, a, s') p(s'|s, a) = 0$. This constraint is equivalent to the $\hat{A}$ constraint in DAE, in the sense that they both ensure the functions satisfy the centering property of the advantage function (i.e., $\mathbb{E}_{a \sim \pi}[A^\pi(s, a)|s] = 0$). Below, we briefly discuss how to deal with this.

**Approximating the constraint** As a first step, we note that a similar constraint $\sum_{a \in \mathcal{A}} \hat{A}(s, a) \pi(a|s) = 0$ can be enforced through the following parametrization $\hat{A}_\theta(s, a) = f_\theta(s, a) - \sum_{a \in \mathcal{A}} f_\theta(s, a) \pi(a|s)$, where $f_\theta$ is the underlying unconstrained function approximator (Wang et al., 2016b). Unfortunately, this technique cannot be applied directly to the $\hat{B}$ constraint, because (1) it requires a sum over the state space, which is typically too large, and (2) the transition function $p(s'|s, a)$ is usually unknown.

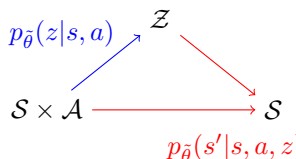

Figure 2: Latent variable model of transitions; $\mathcal{Z}$ is a discrete latent space, which can be understood as actions from nature.

To overcome these difficulties, we use a Conditional Variational Auto-Encoder (CVAE) (Kingma & Welling, 2013; Sohn et al., 2015) to encode transitions into a discrete latent space $\mathcal{Z}$ such that the sum can be efficiently approximated, see Figure 2. The CVAE consists of three components: (1) an approximated conditional posterior $q_{\tilde{\phi}}(z|s, a, s')$ (encoder), (2) a conditional likelihood $p_{\tilde{\theta}}(s'|s, a, z)$ (decoder), and (3) a conditional prior $p_{\tilde{\theta}}(z|s, a)$. These components can then be learned jointly by maximizing the conditional evidence lower bound (ELBO),

$$\text{ELBO} = -D_{\text{KL}}(q_{\tilde{\phi}}(z|s, a, s')||p_{\tilde{\theta}}(z|s, a)) + \mathbb{E}_{z \sim q_{\tilde{\phi}}(\cdot|s, a, s')}[\log p_{\tilde{\theta}}(s'|s, a, z)]. \quad (12)$$

Once a CVAE is learned, we can construct $\hat{B}(s, a, s')$ from an unconstrained function $g_\theta(s, a, z)$ by $B(s, a, s') = \mathbb{E}_{z \sim q_{\tilde{\phi}}(\cdot|s, a, s')}[g_\theta(s, a, z)|s, a, s'] - \mathbb{E}_{z \sim p_{\tilde{\theta}}(\cdot|s, a)}[g_\theta(s, a, z)|s, a]$, which has the property that $\sum_{s'} p(s'|s, a) B(s, a, s') \approx 0$ because $q_{\tilde{\phi}}(z|s, a, s') \approx p_{\tilde{\theta}}(z|s, a, s')$.

## 4 RELATIONSHIP TO OTHER METHODS

In this section, we first demonstrate that (Off-policy) DAE can be understood as a generalization of MC methods with better utilization of trajectories. Secondly, we show that the widely used uncorrected estimator can be seen as a special case of Off-policy DAE and shed light on when it might work.

### 4.1 MONTE-CARLO METHODS

To understand how DAE can speed up learning, let us first revisit Monte-Carlo (MC) methods through the lens of regression. In a typical linear regression problem, we are given a dataset $\{(x_i, y_i) \in \mathbb{R}^n \times \mathbb{R}\}$, and tasked to find coefficients $w \in \mathbb{R}^n$ minimizing the error $\sum_i (w \cdot x_i - y_i)^2$. In RL, the dataset often consists of transitions or sequences of transitions (as in multi-step methods) and their returns, that is, $(\tau_i, G_i)$ where $\tau_i$ has the form $(s_0, a_0, s_1, a_1, ...)$ and $G_i$ is the return associated with $\tau_i$. However, $\tau$ may be an abstract object which cannot be used directly for regression, and

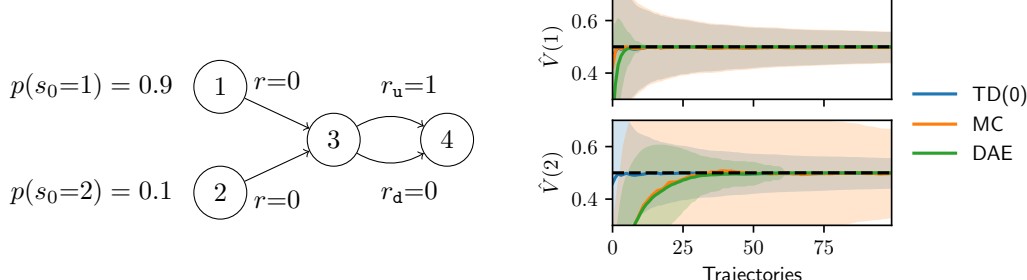

Figure 3: Left: An MDP with $\mathcal{S} = \{1, 2, 3, 4\}$. Both states 1 and 2 have only a single action with immediate rewards 0 that leads to state 3. State 3 has two actions, u and d, that lead to the terminal state 4 with immediate rewards 1 and 0, respectively. Right: We compare the values estimated by Batch TD(0), MC, and DAE with trajectories sampled from the uniform policy. Lines and shadings represent the average and one standard deviation of the estimated values over 1000 random seeds. The dashed line represents the true value $V(1) = V(2) = 0.5$. See Appendix B for details.

we must first map $\tau$ to a feature vector $\phi(\tau) \in \mathbb{R}^n$.[1] For example, in MC methods, we can estimate the value of a state by rolling out trajectories using the target policy starting from the given state and averaging the corresponding returns, i.e., $\mathbb{E}[\sum_{t \geq 0} \gamma^t r_t | s_0 = s] \approx \sum_{i=1}^k G_i / k$. This is equivalent to a linear regression problem, where we first map trajectories to a vector by $\phi_s(\tau) = \mathbb{I}(s_0 = s)$ (vector of length $|\mathcal{S}|$ with elements 1 if the starting state is $s$ or 0 otherwise), and minimize the squared error

$$L(\mathbf{v}) = \sum_{i=1}^k \left[ \left( \sum_s v_s \phi_s(\tau_i) - G_i \right)^2 \right], \tag{13}$$

where $\mathbf{v}$ is the vector of linear regression coefficients $v_s$. Similarly, we can construct feature maps such as $\phi_{s,a}(\tau) = \mathbb{I}(s_0{=}s, a_0{=}a)$ and solve the regression problem to arrive at $Q^\pi(s, a)$. This view shows that MC methods can be seen as linear regression problems with different feature maps. Furthermore, it shows that MC methods utilize rather little information from given trajectories (only the starting state(-action)). An interesting question is whether it is possible to construct features that include more information about the trajectory while retaining the usefulness of the coefficients. Indeed, DAE (Equation 2, with $n \to \infty$) can be seen as utilizing two different feature maps ($\phi_{s,a}(\tau) = \sum_{t=0}^\infty \gamma^t \mathbb{I}(s_t{=}s, a_t{=}a)$ and $\phi_s(\tau) = \mathbb{I}(s_0{=}s)$), which results in a vector of size $|\mathcal{S}| \times |\mathcal{A}|$ that counts the multiplicity of each state-action pair in the trajectory and a vector of size $|\mathcal{S}|$ indicating the starting state. This suggests that DAE can be understood as a generalization of MC methods by using more informative features.

To see how using more informative features can enhance MC methods, let us consider an example (see Figure 3) adapted from Szepesvári (2010). This toy example demonstrates a major drawback of MC methods: it does not utilize the relationship between states 2 and 3, and therefore, an accurate estimate of $\hat{V}(3)$ does not improve the estimate of $\hat{V}(2)$. TD methods, on the other hand, can utilize this relationship to achieve better estimates. DAE, similar to TD methods, also utilizes the relationship between $\hat{V}(2)$ and $\hat{A}(3, \cdot)$ to achieve faster convergence on $\hat{V}(2)$. In fact, in this case, DAE converges even faster than TD(0) as it can exploit the sampling policy to efficiently estimate $\hat{A}(3, \cdot)$, whereas TD(0) has to rely on sample means to estimate $\hat{V}(3)$.

Similarly, we can compare DAE to Off-policy DAE, which further utilizes $\phi_{s,a,s'}(\tau) = \sum_{t=0}^\infty \gamma^t \mathbb{I}(s_t{=}s, a_t{=}a, s_{t+1}{=}s')$, in stochastic environments. See Figure 4 for another example. Here, we observe that both Off-policy DAE variants can outperform DAE even in the on-policy setting. This is because Off-policy DAE can utilize $\hat{B}(4, \cdot, \cdot)$ across different trajectories to account for the variance caused by the stochastic transitions at state 4.

---

[1]This is not to be confused with the features of states, which are commonly used to approximate value functions.

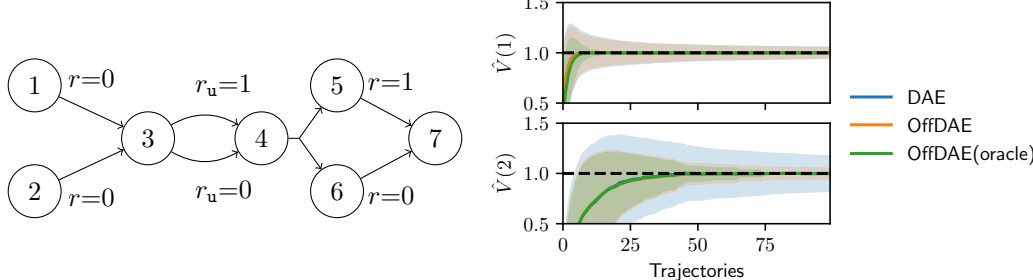

Figure 4: Left: An MDP extended from Figure 3. Instead of terminating at state 4, the agent transitions randomly to state 5 or 6 with equal probabilities. Both states 5 and 6 have a single action, with rewards 1 and 0, respectively. State 7 is the terminal state. Right: We compare the values (with uniform policy) estimated by DAE, Off-policy DAE (learned transition probabilities), and Off-policy DAE (oracle, known transition probabilities). Lines and shadings represent the average and one standard deviation of the estimated values over 1000 random seeds. The dashed line represents the true value $V(1) = V(2) = 1$.

## 4.2 THE UNCORRECTED METHOD

The uncorrected method (simply "Uncorrected" in the following) updates its value estimates using the multi-step target $\sum_{t=0}^{n} \gamma^t r_t + \gamma^{n+1} V_{\text{target}}(s_{n+1})$ without any off-policy correction. Hernandez-Garcia & Sutton (2019) showed that Uncorrected can achieve performance competitive with true off-policy methods in deep RL, although it was also noted that its performance may be problem-specific. Here, we examine how Off-policy DAE, DAE, and Uncorrected relate to each other, and give a possible explanation for when Uncorrected can be used.

We first rewrite the objective of Off-policy DAE (Equation 11) into the following form:

$$\left(\hat{V}(s_0) - \Big(\underbrace{\underbrace{\sum_{t=0}^{n} \gamma^t r_t + \gamma^{n+1} V_{\text{target}}(s_{n+1})}_{\text{Uncorrected}} - \sum_{t=0}^{n} \gamma^t \hat{A}_t}_{\text{DAE}} - \sum_{t=0}^{n} \gamma^{t+1} \hat{B}_t\Big)\right)^2, \tag{14}$$

where the underbraces indicate the updating targets of the respective method. We can see now there is a clear hierarchy between these methods, where DAE is a special case of Off-policy DAE by assuming $\hat{B} \equiv 0$, and Uncorrected is a special case by assuming both $\hat{A} \equiv 0$ and $\hat{B} \equiv 0$.

The question is, then, when is $\hat{A} \equiv 0$ or $\hat{B} \equiv 0$ a good assumption? Remember that, in deterministic environments, we have $B^\pi \equiv 0$ for any policy $\pi$; therefore, $\hat{B} \equiv 0$ is a correct estimate of $B^\pi$, meaning that DAE can be directly applied to off-policy data when the environment is deterministic. Next, to see when $\hat{A} \equiv 0$ is useful, remember that the advantage function can be interpreted as the causal effect of an action on the return. In other words, if actions in the environment tend to have minuscule impacts on the return, then Uncorrected can work with a carefully chosen backup length. This can partially explain why Uncorrected worked in environments like Atari games (Bellemare et al., 2013; Gruslys et al., 2017; Hessel et al., 2018) for small backup lengths, because the actions are fine-grained and have small impact ($A \approx 0$) in general. In Appendix C, we provide a concrete example demonstrating how ignoring the correction can lead to biased results.

## 5 EXPERIMENTS

We now compare (1) Uncorrected, (2) DAE, (3) Off-policy DAE, and (4) Tree Backup (Precup et al., 2000) in terms of policy optimization performance using a simple off-policy actor-critic algorithm. By comparing (1), (2), and (3), we test the importance of $\hat{A}$ and $\hat{B}$ as discussed in Section 4.2. Method (4) serves as a baseline of true off-policy method, and Tree Backup was chosen because,

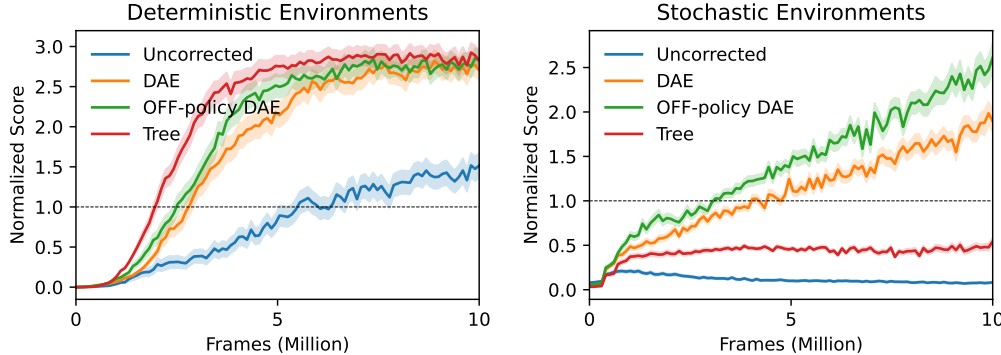

Figure 5: Normalized training curves aggregated over deterministic (left) and stochastic (right) environments. Scores were first normalized using the PPO-DAE baseline and then aggregated over 20 random seeds, environments, and backup lengths. Lines and shadings represent the means and 1 standard error of the means, respectively. The dotted horizontal lines shows the PPO-DAE baseline.

like Off-policy DAE, it also assumes no knowledge of the behavior policy, in contrast to importance sampling methods. We compare these methods in a controlled setting, by only changing the critic objective with all other hyperparameters fixed.

**Environment** We perform our experiments using the MinAtar suite (Young & Tian, 2019). The MinAtar suite consists of 5 environments that replicate the dynamics of a subset of environments from the Arcade Learning Environment (ALE) (Bellemare et al., 2013) with simplified state/action spaces. The MinAtar environments have several properties that are desirable for our study: (1) Actions tend to have significant consequences due to the coarse discretization of its state/action spaces. This suggests that ignoring other actions' effects ($\hat{A}$), as done in Uncorrected, may have a larger impact on its performance. (2) The MinAtar suite includes both deterministic and stochastic environments, which allows us to probe the importance of $\hat{B}$.

**Agent Design** We summarize the agent in Algorithm 1. Since (Off-policy) DAE's loss function depends heavily on the target policy, we found that having a smoothly changing target policy during training is critical, especially when the backup length is long. Preliminary experiments indicated that using the greedy policy, i.e., $\arg\max_{a \in \mathcal{A}} \hat{A}(s, a)$, as the target policy can lead to divergence, which is likely due to the phenomenon of policy churning (Schaul et al., 2022). To mitigate this, we distill a policy by maximizing $\mathbb{E}_{a \sim \pi_\theta}[\hat{A}(s, a)]$, and smooth it with exponential moving average (EMA). The smoothed policy $\pi_{\mathrm{EMA}}$ is then used as the target policy. Additionally, to avoid premature convergence, we include a KL-divergence penalty between $\pi_\theta$ and $\pi_{\mathrm{EMA}}$, similar to trust-region methods (Schulman et al., 2015a). For critic training, we also use an EMA of past value functions as the bootstrapping target. For Off-policy DAE, we additionally learn a CVAE model of the environment. Since learning

---

**Algorithm 1** A Simple Actor-Critic Algorithm

**Require:** `backup`
1: Initialize $A_\theta, V_\theta, B_\theta, \pi_\theta$
2: Initialize CVAE $q_{\tilde{\phi}}, p_{\tilde{\theta}}$
3: Initialize $D \leftarrow \{\}, \theta_{\mathrm{EMA}} \leftarrow \theta$
4: **for** $t = 0, 1, 2, \dots$ **do**
5:     Sample $(s, a, r, s') \sim \pi_\theta$
6:     $D \leftarrow D \cup \{(s, a, r, s')\}$
7:     Sample batch $\mathcal{B}$ trajectories from $D$
8:     **if** `backup` is Off-policy DAE **then**
9:         Train CVAE (Eq 12) using $\mathcal{B}$
10:         Approximate $B_\theta(s, a, s')$
11:     **end if**
12:     Compute $L_{\mathrm{critic}}$ (Eq. 14)
13:     Compute $L_{\mathrm{actor}}$
        $= -\mathbb{E}_{a \sim \pi_\theta}[\hat{A}] + \beta_{\mathrm{KL}} D_{\mathrm{KL}}(\pi_\theta || \pi_{\theta_{\mathrm{EMA}}})$
14:     Train $L_{\mathrm{critic}} + L_{\mathrm{actor}}$ using $\mathcal{B}$
15:     $\theta_{\mathrm{EMA}} \leftarrow \tau \theta_{\mathrm{EMA}} + (1 - \tau)\theta$
16: **end for**

---

the dynamics of the environment may improve sample efficiency by learning better representations (Gelada et al., 2019; Schwarzer et al., 2020; Hafner et al., 2020), we isolate this effect by training a separate network for the CVAE such that the agent can only query $p(z|s, a, s')$ and $p(z|s, a)$. See Appendix D for more details about the algorithm and hyperparameters.

**Results** Each agent is trained for 10 million frames, and evaluated by averaging the undiscounted scores of 100 episodes obtained by the trained policy. For comparison, we use the scores reported by Pan et al. (2022) as an on-policy baseline, which were trained using PPO and DAE (denoted PPO-DAE). The results are summarized in Figure 5. Additional results for individual environments and other ablation studies can be found in Appendix D. We make the following observations: (1) For deterministic environments, both DAE variants performed similarly, demonstrating that $\hat{B}$ is irrelevant. Additionally, both DAE variants converged to similar scores as Tree backup, albeit slightly slower, suggesting that they can compete with true off-policy methods. Uncorrected, on the other hand, performed significantly worse than DAE, suggesting that $\hat{A}$ is crucial in off-policy settings, as the two methods only differ in $\hat{A}$. (2) For stochastic environments, we see a clear hierarchy between Uncorrected, DAE and Off-policy DAE, suggesting that both $\hat{A}$ and $\hat{B}$ corrections are important. Notably, Tree backup performs significantly worse than both DAE variants in this case, while only being slightly better than Uncorrected.

## 6 RELATED WORK

**Advantage Function** The advantage function was originally proposed by Baird (1994) to address small time-step domains. Later, it was shown that the advantage function can be used to relate value functions of different policies (Kakade & Langford, 2002) or reduce the variance of policy gradient methods (Greensmith et al., 2004). These properties led to wide adoption of the advantage function in modern policy optimization methods (Schulman et al., 2015a;b; 2017; Mnih et al., 2016). More recently, the connection between causal effects and the advantage function was pointed out by Corcoll & Vicente (2020), and further studied by Pan et al. (2022), who also proposed DAE.

**Multi-step Learning** Multi-step methods (Watkins, 1989; Sutton, 1988) have been widely adopted in recent deep RL research and shown to have a strong effect on performance (Schulman et al., 2015b; Hessel et al., 2018; Wang et al., 2016a; Gruslys et al., 2017; Espeholt et al., 2018; Hernandez-Garcia & Sutton, 2019). Typical off-policy multi-step methods include importance sampling (Munos et al., 2016; Rowland et al., 2020; Precup et al., 2001), truncating (diminishing) off-policy actions (Watkins, 1989; Precup et al., 2000), a combination of the two (De Asis et al., 2018), or simply ignoring any correction.

**Afterstates** The idea of dissecting transitions into a two-step process dates at least back to Sutton et al. (1998), where afterstates (equivalent to nature's states in Figure 1) were introduced. It was shown that learning the values of afterstates can be easier in some problems. Similar ideas also appeared in the treatment of random events in extensive-form games, where they are sometimes referred to as "move by nature" (Fudenberg & Tirole, 1991).

**Luck** Mesnard et al. (2021) proposed to use future-conditional value functions to capture the effect of luck, and demonstrated that these functions can be used as baselines in policy gradient methods to reduce variance. In this work, we approached this problem from a causal effect perspective and provided a quantitative definition of luck (see Equation 10).

## 7 DISCUSSION

In the present work, we demonstrated how DAE can be extended to off-policy settings. We also relate Off-policy DAE to previous methods to better understand how it can speed up learning. Through experiments in both stochastic and deterministic environments, we verified that the proposed off-policy correction is beneficial for policy optimization.

One limitation of the proposed method lies in enforcing the $\hat{B}$ constraint in stochastic environments. In the present work, this was approximated using CVAEs, which introduced computational overhead and additional hyperparameters. One way to reduce computational overhead and scale to high dimensional domains is to learn a value equivalent model (Antonoglou et al., 2021; Grimm et al., 2020). We will leave it as future work to explore more efficient ways to enforce the constraint.

## ACKNOWLEDGMENTS

HRP would like to thank Nico Gürtler for the constructive feedback. The authors thank the International Max Planck Research School for Intelligent Systems (IMPRS-IS) for supporting Hsiao-Ru Pan.

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

## A   PROOF OF THEOREM 1

**Theorem** (Off-policy DAE). *Given a behavior policy $\mu$ and a target policy $\pi$ and backup length $n \geq 0$. Let $\hat{A}_t = \hat{A}(s_t, a_t)$, $\hat{B}_t = \hat{B}(s_t, a_t, s_{t+1})$, and the constrained squared error*

$$L(\hat{A}, \hat{B}, \hat{V}) = \mathbb{E}_{\mu}\left[\left(\sum_{t=0}^{n} \gamma^t\left(r_t - \hat{A}_t - \gamma\hat{B}_t\right) + \gamma^{n+1}\hat{V}(s_{n+1}) - \hat{V}(s_0)\right)^2\right] \tag{15}$$

$$\text{subject to} \begin{cases} \sum_{a\in\mathcal{A}} \hat{A}(s,a)\pi(a|s) = 0 & \forall s \in \mathcal{S} \\ \sum_{s'\in\mathcal{S}} \hat{B}(s,a,s')p(s'|s,a) = 0 & \forall(s,a) \in \mathcal{S}\times\mathcal{A} \end{cases},$$

*then $(A^\pi, B^\pi, V^\pi)$ is a minimizer of the above problem. Furthermore, the minimizer is unique if $\mu$ is sufficiently explorative (i.e., non-zero probability of reaching all possible transitions $(s, a, s')$).*

*Proof.* Since

$$0 \leq L(A^\pi, B^\pi, V^\pi) = \mathbb{E}_{\mu}\left[\left(\sum_{t=0}^{n} \gamma^t(r_t - A_t^\pi - \gamma B_t^\pi) + \gamma^{n+1}V^\pi(s_{n+1}) - V^\pi(s_0)\right)^2\right] = 0, \tag{16}$$

and both $\sum_{a\in\mathcal{A}} \pi(a|s)A^\pi(s,a) = 0$ and $\sum_{s'\in\mathcal{S}} p(s'|s,a)B^\pi(s,a,s') = 0$ constraints are satisfied, $(A^\pi, B^\pi, V^\pi)$ is a minimizer of the problem. For uniqueness, we assume the behavior policy is sufficiently explorative such that any sequence $(s_0, a_0, r_0, ...s_{n+1})$ has non-zero probability of being visited. Now, suppose there exists $(A', B', V')$ that also minimizes $L$, i.e., $L(A', B', V') = 0$, then for any sequence $(s_0, a_0, ...s_{t+1})$, we must have

$$\sum_{t=0}^{n} \gamma^t(r_t - A'_t - \gamma B'_t) + \gamma^{n+1}V'(s_{n+1}) - V'(s_0) = 0, \tag{17}$$

otherwise $L(A', B', V') \neq 0$. If we take the conditional expectation over $(a_0, s_1, a_1, ...s_{n+1})$ conditioned on $s_0$ using the target policy $\pi$, then

$$V'(s_0) = \mathbb{E}_{\pi}[\sum_{t=0}^{n} \gamma^t(r_t - A'_t - \gamma B'_t) + \gamma^{n+1}V'(s_{n+1})|s_0] \tag{18}$$

$$= \mathbb{E}_{\pi}[\sum_{t=0}^{n} \gamma^t r_t + \gamma^{n+1}V'(s_{n+1})|s_0], \tag{19}$$

which means that $V'$ satisfies the Bellman Equation. Therefore $V' = V^\pi$ uniquely. Similarly, if we take the expectation over $(s_1, a_1, ...s_{n+1})$ conditioned on $s_0, a_0$, then

$$A'(s_0, a_0) = r(s_0, a_0) + \mathbb{E}_{\pi}[\sum_{t=1}^{n} \gamma^t r_t + \gamma^{n+1}V^\pi(s_{n+1})|s_0, a_0] - V^\pi(s_0) = A^\pi(s_0, a_0) \tag{20}$$

Finally, if we take the expectation over $(a_1, s_2, ..., s_{t+1})$ conditioned on $s_0, a_0, s_1$, then

$$\gamma B'(s_0, a_0, s_1) = r(s_0, a_0) - A^\pi(s_0, a_0) + \mathbb{E}_{\pi}[\sum_{t=1}^{n} \gamma^{t'} r_{t'} + \gamma^{t+1}V^\pi(s_{t+1})|s_0, a_0, s_1] - V^\pi(s_0) \tag{21}$$

$$= \gamma(V^\pi(s_1) - \mathbb{E}[V^\pi(s_1)|s_0, a_0]) = \gamma B^\pi(s_0, a_0, s_1). \tag{22}$$

Similarly, we get $(A', B', V') = (A^\pi, B^\pi, V^\pi)$ for all $(s_{t'}, a_{t'}, s_{t'+1})$ with $0 \leq t' \leq n$ by repeatedly taking the conditional expectations over the sequence. By the assumption that $\mu$ has non-zero probability of visiting any sequence, we have $(A', B', V') = (A^\pi, B^\pi, V^\pi)$ for all $(s, a, s') \in \mathcal{S} \times \mathcal{A} \times \mathcal{S}$. □

Remarks: (1) While we used the squared error as the objective function in the theorem, it can be replaced with an arbitrary metric in $\mathbb{R}$, as the proof does not rely on properties of the squared error. (2) For uniqueness, the condition on the behavior policy $\mu$ can be relaxed if we only care about states/actions covered by the target policy $\pi$. In that case, we only need the coverage of $\mu$ to include the coverage of $\pi$.

# B DETAILS OF FIGURE 3 AND FIGURE 4

In this example, we compare the sample efficiency of MC, Batch TD(0) and DAE. We note that there are only 4 possible trajectories in this environment, depending on the starting state (1 or 2) and the action chosen at state 3 (u or d). We denote the number of trajectories starting from $i$ and choosing action $a \in \{u, d\}$ by $n_{i,a}$, and $n_i = n_{i,u} + n_{i,d}$. The trajectories were sampled using the uniform policy, i.e., $\pi(u|3) = \pi(d|3) = 0.5$. In the following list, we summarize the estimates from each method.

- MC: $\hat{V}(1) = \frac{n_{1,u}}{n_1}, \quad \hat{V}(2) = \frac{n_{2,u}}{n_2}$

- Batch TD(0): $\hat{V}(1) = \hat{V}(2) = \hat{V}(3) = \frac{n_{1,u}+n_{2,u}}{n_1+n_2}$

- DAE: The minimizer of

$$L(\hat{V}(1), \hat{V}(2), \hat{A}(3, u), \hat{A}(3, d)) = n_{1,u}(1 - \hat{A}(3, u) - \hat{V}(1))^2 + n_{1,d}(0 - \hat{A}(3, d) - \hat{V}(1))^2$$
$$+ n_{2,u}(1 - \hat{A}(3, u) - \hat{V}(2))^2 + n_{2,d}(0 - \hat{A}(3, d) - \hat{V}(2))^2$$

subject to $\hat{A}(3, u) + \hat{A}(3, d) = 0$ (since the sampling policy is uniform).

One can use the method of Lagrange multiplier to solve the DAE problem and arrive at the following linear equations:

$$\begin{pmatrix} n_{1,u} - n_{1,d} & n_1 & 0 \\ n_{2,u} - n_{2,d} & 0 & n_2 \\ (n_{1,u} + n_{2,u}) & n_{1,u} & n_{2,u} \end{pmatrix} \begin{pmatrix} \hat{A}(3, u) \\ \hat{V}(1) \\ \hat{V}(2) \end{pmatrix} = \begin{pmatrix} n_{1,u} \\ n_{2,u} \\ n_{1,u} + n_{2,u} \end{pmatrix}, \tag{23}$$

Note that there are only 3 equations, since $\hat{A}(3, u) + \hat{A}(3, d) = 0$. Additionally, the solution is unique only when both $n_1 > 0$ and $n_2 > 0$, otherwise the first row or the second row of the matrix would be 0. For simplicity, we use the pseudoinverse to compute the solution:

$$\begin{pmatrix} \hat{A}(3, u) \\ \hat{V}(1) \\ \hat{V}(2) \end{pmatrix} = \begin{pmatrix} n_{1,u} - n_{1,d} & n_1 & 0 \\ n_{2,u} - n_{2,d} & 0 & n_2 \\ (n_{1,u} + n_{2,u}) & n_{1,u} & n_{2,u} \end{pmatrix}^+ \begin{pmatrix} n_{1,u} \\ n_{2,u} \\ n_{1,u} + n_{2,u} \end{pmatrix}, \tag{24}$$

where $+$ denotes the pseudoinverse. This explains why the DAE estimates in Figure 3 are slightly skewed towards 0 at the beginning. Figure 4 can be obtained similarly by adding the $\hat{B}$ terms to the DAE loss. One difference is that, in the case of learned transition probabilities, the $\hat{B}$ constraint was enforced based on the estimated transition probabilities instead of a fixed distribution.

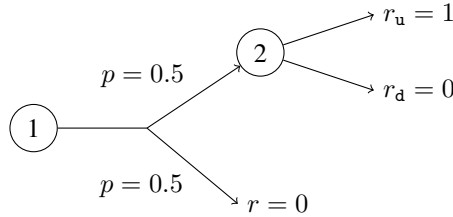

Figure 6: A simple MDP with $\mathcal{S} = \{1, 2\}$. State 1 has only a single action, which can lead to state 2 or the terminal state with equal probabilities. State 2 has two actions $\mathtt{u}$ and $\mathtt{d}$ with rewards 1 and 0, respectively, and both actions lead to the terminal state.

## C    COUNTEREXAMPLE

In this section, we construct an example to demonstrate that naively applying DAE to off-policy data can lead to biased results. Consider the environment in Figure 6. Suppose the data is collected with a behavior policy $\mu$ and we wish to estimate values for a target policy $\pi$. If we apply DAE directly to this problem without any off-policy correction, then the loss is equal to

$$L(\hat{A}, \hat{V}) = \frac{1}{2}\mu(\mathtt{u}|2) \left(1 - \hat{A}(2, \mathtt{u}) - \hat{V}(1)\right)^2 + \frac{1}{2}\mu(\mathtt{d}|2) \left(0 - \hat{A}(2, \mathtt{d}) - \hat{V}(1)\right)^2 + \frac{1}{2}(0 - \hat{V}(1))^2,$$
(25)

since there are only three possible trajectories in this environment. Now, if we include the constraint that $\sum_a \hat{A}(2, a)\pi(a|2) = 0$, then the problem can be solved by the method of Lagrange multiplier using the following Lagrangian:

$$L(\hat{A}, \hat{V}) + \lambda \sum_a \hat{A}(2, a)\pi(a|2).$$
(26)

The minimizer $(A^*, V^*) = \arg\min L(\hat{A}, \hat{V})$ is given by:

$$\begin{cases} V^*(1) = \frac{\pi(\mathtt{u}|2)}{1 + \frac{\pi(\mathtt{u}|2)^2}{\mu(\mathtt{u}|2)} + \frac{\pi(\mathtt{d}|2)^2}{\mu(\mathtt{d}|2)}} \\ \lambda = V^*(1) \\ A^*(2, \mathtt{u}) = 1 - V^*(1) - \frac{V^*(1)\pi(\mathtt{u}|2)}{\mu(\mathtt{u}|2)} \\ A^*(2, \mathtt{d}) = 0 - V^*(1) - \frac{V^*(1)\pi(\mathtt{d}|2)}{\mu(\mathtt{d}|2)} \end{cases}$$
(27)

meaning that $V^*(1) \neq V^\pi(1) = \frac{\pi(\mathtt{u}|2)}{2}$, and $A^* \neq A^\pi$ if $\pi \neq \mu$. One can also verify that, if $\pi = \mu$ (on-policy), then $(V^*, A^*) = (V^\pi, A^\pi)$.

# D    DETAILS OF THE MINATAR EXPERIMENTS

## D.1    PSEUDOCODE

The more detailed pseudocode of the proposed actor-critic method is provided in Algorithm 2. Here we note some details about the training process. Unlike typical methods that store 1-step transitions in the replay buffer, our buffer consists of $n$-step trajectories. When computing the critic loss, we also compute the loss for each sub-trajectory as in Pan et al. (2022). For example, if $(s_0, a_0, ..., s_n)$ is a sample trajectory, we accumulate the critic loss for all sub-trajectory $(s_i, a_i, ..., s_n)$ for all $i \in \{0, 1, 2, ..., n-1\}$. Also, to speed up training, we use parallel actors to sample transitions from the environments.

---

**Algorithm 2** A simple Off-policy Actor-Critic Algorithm

---

**Require:** `backup` $\in$ {Uncorrected, DAE, Off-policy DAE, Tree}
**Require:** $n$ (backup length)
1: Initialize actor-critic components $A_\theta(s, a)$, $V_\theta(s)$, $B_\theta(s, a, z)$, $\pi_\theta(a|s)$
2: Initialize CVAE $q_{\tilde\phi}(z|s, a, s')$, $p_{\tilde\theta}(z|s, a)$, $p_{\tilde\theta}(s'|s, a, z)$
3: $\theta_{\text{EMA}} \leftarrow \theta$
4: $D = \{\}$
5: $D_n = \{\}$
6: **for** $t = 0, 1, 2, \ldots$ **do**
7:     Sample transition $(s, a, r, s')$ with policy $\pi_\theta$
8:     $D_n \leftarrow D_n \cup \{(s, a, r, s')\}$
9:     **if** $s'$ is terminal **or** $|D_{\text{n-step}}| = n$ **then**
10:         $D \leftarrow D \cup \{\texttt{concatenate}(D_n)\}$
11:         $D_n \leftarrow \{\}$
12:     **end if**
13:     **if** $t + 1 \bmod$ `steps_per_update` $= 0$ **then**
14:         Sample batch of trajectories $\mathcal{B}$ from $D$
15:         **if** `backup` = Off-policy DAE **then**
16:             Train $q_{\tilde\phi}(z|s, a, s')$, $p_{\tilde\theta}(z|s, a)$, $p_{\tilde\theta}(s'|s, a, z)$ using $\mathcal{B}$ by Equation 12
17:             $B_\theta(s, a, s') \leftarrow \mathbb{E}_{z \sim q_{\tilde\phi}(\cdot|s,a,s')}[B_\theta(s, a, z)|s, a, s'] - \mathbb{E}_{z \sim p_{\tilde\theta}(\cdot|s,a)}[B_\theta(s, a, z)|s, a]$
18:         **end if**
19:         $A_\theta(s, a) \leftarrow A_\theta(s, a) - \mathbb{E}_{a \sim \pi_{\theta_{\text{EMA}}}}[A_\theta(s, a)]$
20:         Compute critic loss $L_{\text{critic}}$ according to `backup` (Eq. 14 or Eq. 28)
21:         $A_{\text{normalized}} \leftarrow \texttt{stop\_gradient}(A_\theta(s, a)/\sqrt{\mathbf{Var}[A_\theta]})$
22:         Compute actor loss $L_{\text{actor}} = -\mathbb{E}_{a \sim \pi_\theta}[A_{\text{normalized}}] + \beta_{\text{KL}} D_{\text{KL}}(\pi_\theta||\pi_{\theta_{\text{EMA}}})$
23:         Train $L_{\text{critic}} + L_{\text{actor}}$ with Adam (Kingma & Ba, 2014)
24:         $\theta_{\text{EMA}} \leftarrow \tau\theta_{\text{EMA}} + (1 - \tau)\theta$
25:     **end if**
26: **end for**

---

The $n$-step Tree backup Q-target is defined recursively by:

$$Q_{\text{target}}^n(s_t, a_t) = r(s_t, a_t) + \gamma \left( \sum_{a \neq a_{t+1}} \pi(a|s_{t+1})Q_{\text{target}}(s_{t+1}, a) + \pi(a_{t+1}|s_{t+1})Q_{\text{target}}^{n-1}(s_{t+1}, a_{t+1}) \right) \tag{28}$$

where $Q_{\text{target}} = V_{\theta_{\text{EMA}}} + A_{\theta_{\text{EMA}}}$ in our case.

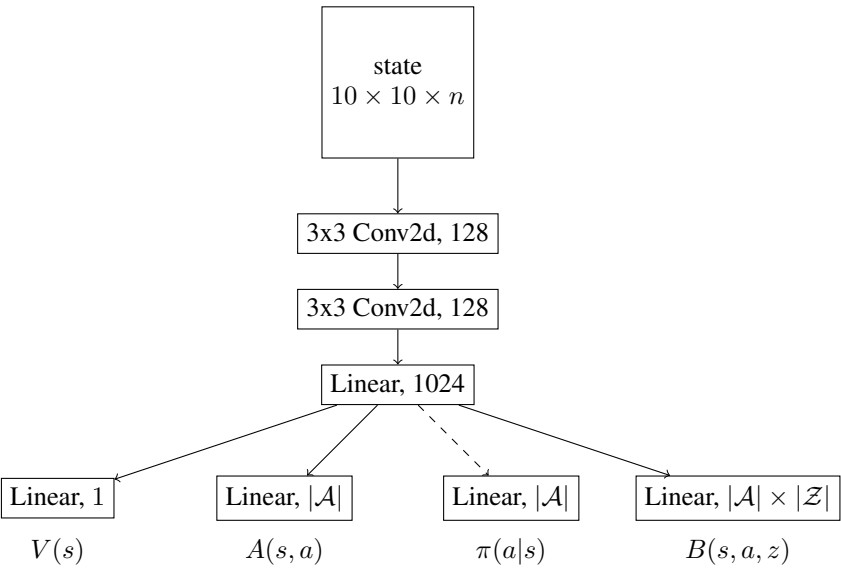

Figure 7: Network architecture for the actor-critic. Each hidden layer is followed by a ReLU activation function. The dashed line indicates that gradients are stopped.

## D.2 ACTOR-CRITIC NETWORK

In our experiments, we use a convolutional neural network followed by multiple heads to approximate $A_\theta$, $B_\theta$, $V_\theta$, and $\pi_\theta$ (see Figure 7)(Mnih et al., 2016; Wang et al., 2016b). Since we train both the actor and the critic using a single network simultaneously, to avoid interference between the two losses ($L_{\text{critic}}$ and $L_{\text{actor}}$), we simply use the representation learned from by the critic to train the actor by stopping the gradients from $L_{\text{actor}}$ to the shared network. This eliminates the need to balance for the different loss functions.

## D.3 CONDITIONAL VARIATIONAL AUTOENCODER (CVAE) NETWORK

We illustrate the training process and the network architecture in Figure 8. First, we use a small discrete latent space $\mathcal{Z}$ which allows us to compute expectations over $\mathcal{Z}$ efficiently. This also alleviates the need to use heuristics such as VQ-VAE (van den Oord et al., 2017) or Gumbel-Softmax (Jang et al., 2016; Maddison et al., 2016), because we can now compute $\mathbb{E}_{z \sim q_{\tilde{\phi}}(\cdot|s,a,s')}[\log p_{\tilde{\theta}}(s'|s,a,z)] = \sum_{z \in \mathcal{Z}} q_{\tilde{\phi}}(z|s,a,s') \log p_{\tilde{\theta}}(s'|s,a,z)$ exactly. Second, to eliminate the need to balance between KL-divergence loss and the reconstruction loss, the conditional prior is trained using the representation from the encoder with gradients stopped. This is similar to the approach in VQ-VAE where a prior is trained separately. Finally, we observed that the posterior can sometimes collapse early in training. To mitigate this, we add a small entropy penalty for the posterior. Combining everything together, we have the loss function for CVAE:

$$\mathcal{L}_{\text{CVAE}}(\tilde{\theta}, \tilde{\phi}; s, a, s') = D_{\text{KL}}(q_{\tilde{\phi}}(z|s,a,s')||p_{\tilde{\theta}}(z|s,a)) - \mathbb{E}_{z \sim q_{\tilde{\phi}}(\cdot|s,a,s')}[\log p_{\tilde{\theta}}(s'|s,a,z)]$$
$$-\beta_{\text{ent}}H(q_{\tilde{\phi}}(\cdot|s,a,s'))$$

where $H(\cdot)$ is the entropy, and $\beta_{\text{ent}}$ controls the strength of the entropy penalty.

## D.4 HYPERPARAMETERS

In Table 1, we summarize the hyperparameters used in the MinAtar experiments. In general, the agent was designed to have very few hyperparameters to reduce potential confounding when comparing different backup methods. Our preliminary experiments found that the effects of the hyperparameters to be agnostic to backup methods, except for $\tau$ which tend to have a larger impact on Off-policy DAE and DAE, which is likely due to the heavy dependence on the policy when training with DAE-like loss functions.

For CVAE training, we found the Adam's suggested $\beta = (0.9, 0.999)$ can sometimes lead to divergence, and lowering it to $(0.5, 0.9)$ can greatly improve stability.

Table 1: List of hyperparameters. Note that for Off-policy DAE, there are two separate optimizers used for actor-critic and CVAE training. $^{\dagger}$: not used in Pan et al. (2022).

| Group | Parameter | Value |
|---|---|---|
| Environment setting | Sticky Action | False |
| | Difficulty Ramping | False |
| | Maximum Episode Length$^{\dagger}$ | 108000 frames |
| Shared (actor-critic training) | Discount $\gamma$ | 0.99 |
| | Parallel actors | 128 |
| | Initial steps before training | 25000 frames |
| | Replay Buffer Size | 1000000 frames |
| | Backup Length | 8/16/32 |
| | Optimizer | Adam(Kingma & Ba, 2014) |
| | Learning rate | 0.00025 (linearly annealed to 0) |
| | Adam $\beta$ | (0.9, 0.999) |
| | Adam $\epsilon$ | $10^{-4}$ |
| | Env. steps per update | 32 |
| | Batch Size | 1024 frames |
| | $\beta_{\text{KL}}$ | 3.0 |
| | $\tau$ (EMA weight) | 0.999 |
| Off-policy DAE only (CVAE training) | Latent size $|\mathcal{Z}|$ | 16 |
| | Optimizer | Adam |
| | Learning rate | 0.00025 |
| | Adam $\beta$ | (0.5, 0.9) |
| | Adam $\epsilon$ | $10^{-8}$ |
| | $\beta_{\text{ent}}$ | 0.0001 |

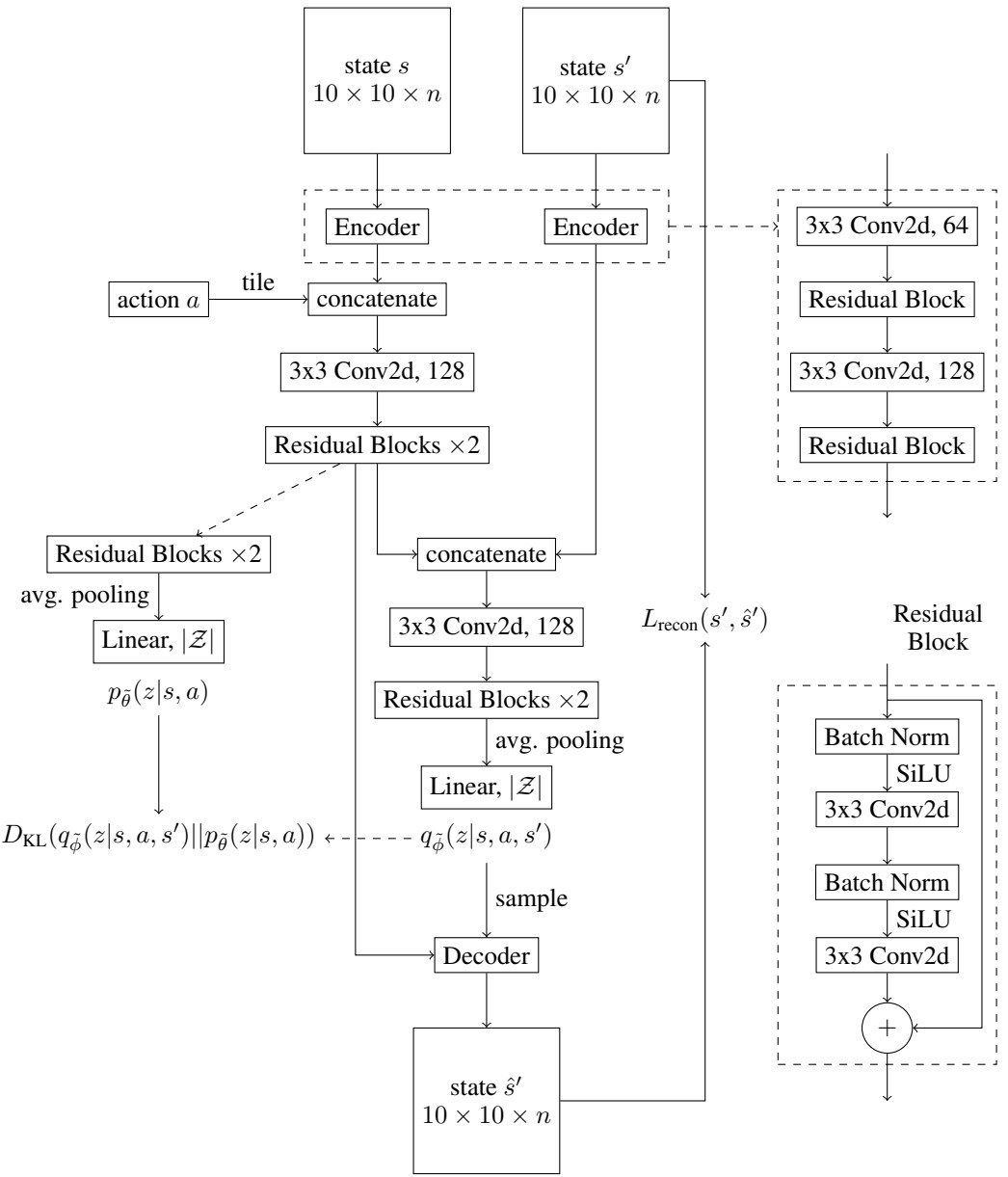

Figure 8: Network architecture and the training graph for the CVAE. Dashed lines indicate that gradients are stopped. The Decoder uses the same architecture as the encoder with orders reversed and convolutions replaced with transposed convolutions. Softmax is applied to both $q_{\tilde{\phi}}(z|s,a,s')$ and $p_{\tilde{\theta}}(z|s,a)$ to ensure they are probability distributions over $\mathcal{Z}$ (note the $Z$ is a discrete space in this case). We use binary cross entropy for the reconstruction loss, since states in the MinAtar environments are binary images.

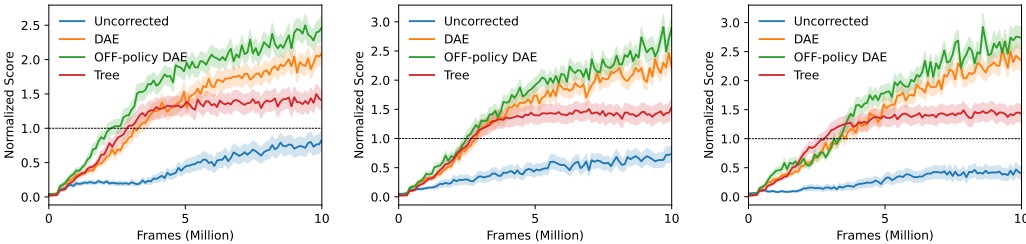

Figure 9: Normalized training curves for each backup method and backup length (from left to right: 8, 16, 32). The dashed horizontal line represents the PPO-DAE baseline.

## D.5 MORE RESULTS

In the MinAtar experiments, normalized score were calculated by $\text{score}_{\text{norm}} = \frac{\text{score}-\text{baseline}}{\text{baseline}}$, where we use the scores reported by Pan et al. (2022) as baselines. We note that in the MinAtar suite, Breakout[2] and Space Invaders are deterministic environments, while Asterix, Freeway, and Seaquest are stochastic environments. Interestingly, we found that Off-policy DAE tends to slightly outperform DAE on Space Invaders (see Figure 11). This is likely because Space Invaders is partially observable, and modeling the transitions as stochastic can be helpful.

Table 2: Score comparison between different methods and backup lengths. Results were aggregated over 20 random seeds. Numbers represent (mean)±(1 standard error).

| Environment | N | Backup Method | | | |
| --- | --- | --- | --- | --- | --- |
| | | Uncorrected | DAE | Off-policy DAE | Tree |
| Asterix | 8 | $4.5 \pm 0.2$ | $155.6 \pm 5.4$ | $161.5 \pm 5.9$ | $44.3 \pm 1.1$ |
| | 16 | $2.5 \pm 0.1$ | $194.2 \pm 4.8$ | $207.0 \pm 9.2$ | $39.0 \pm 1.4$ |
| | 32 | $2.0 \pm 0.1$ | $215.4 \pm 8.1$ | $268.9 \pm 14.0$ | $39.0 \pm 1.3$ |
| Breakout | 8 | $5799.9 \pm 611.3$ | $9573.0 \pm 250.1$ | $9423.3 \pm 351.9$ | $9411.4 \pm 505.5$ |
| | 16 | $5220.1 \pm 448.7$ | $8506.1 \pm 476.6$ | $8887.9 \pm 429.0$ | $10069.0 \pm 288.8$ |
| | 32 | $3179.3 \pm 661.8$ | $8119.8 \pm 617.5$ | $7372.1 \pm 582.0$ | $10139.8 \pm 338.1$ |
| Freeway | 8 | $5.5 \pm 1.8$ | $55.1 \pm 0.1$ | $61.9 \pm 0.6$ | $2.2 \pm 0.4$ |
| | 16 | $16.8 \pm 1.3$ | $57.6 \pm 0.1$ | $62.3 \pm 0.3$ | $4.1 \pm 1.0$ |
| | 32 | $8.2 \pm 0.8$ | $58.8 \pm 0.1$ | $62.9 \pm 0.3$ | $5.1 \pm 0.8$ |
| Seaquest | 8 | $13.5 \pm 1.5$ | $413.5 \pm 25.1$ | $839.4 \pm 48.3$ | $312.3 \pm 16.4$ |
| | 16 | $4.1 \pm 0.1$ | $594.3 \pm 36.3$ | $1171.6 \pm 66.6$ | $286.4 \pm 11.6$ |
| | 32 | $4.0 \pm 0.1$ | $821.6 \pm 52.5$ | $1225.3 \pm 63.7$ | $266.2 \pm 19.1$ |
| SpaceInvaders | 8 | $5561.5 \pm 452.3$ | $15116.1 \pm 377.4$ | $18086.9 \pm 419.1$ | $15615.3 \pm 563.0$ |
| | 16 | $1180.3 \pm 145.0$ | $13478.7 \pm 391.2$ | $16756.8 \pm 460.4$ | $16009.6 \pm 508.1$ |
| | 32 | $307.7 \pm 32.0$ | $12560.4 \pm 441.4$ | $14970.3 \pm 348.9$ | $17374.2 \pm 584.9$ |

## D.6 WITH AND WITHOUT TARGET NETWORK

While Equation 11 suggests that a separate target network for critic training is not necessary, in practice, we have found that using a target network leads to better performance. See Figure 12 for results regarding the effect of target networks. In general, we found the critic loss to be lower when using target networks. One possible explanation is that using target networks results in biased, but lower variance estimates, which in turn makes the loss easier to optimize. Further investigation is required to understand the tradeoffs.

---

[2]The initial states in Breakout are random, but the transitions are stochastic.

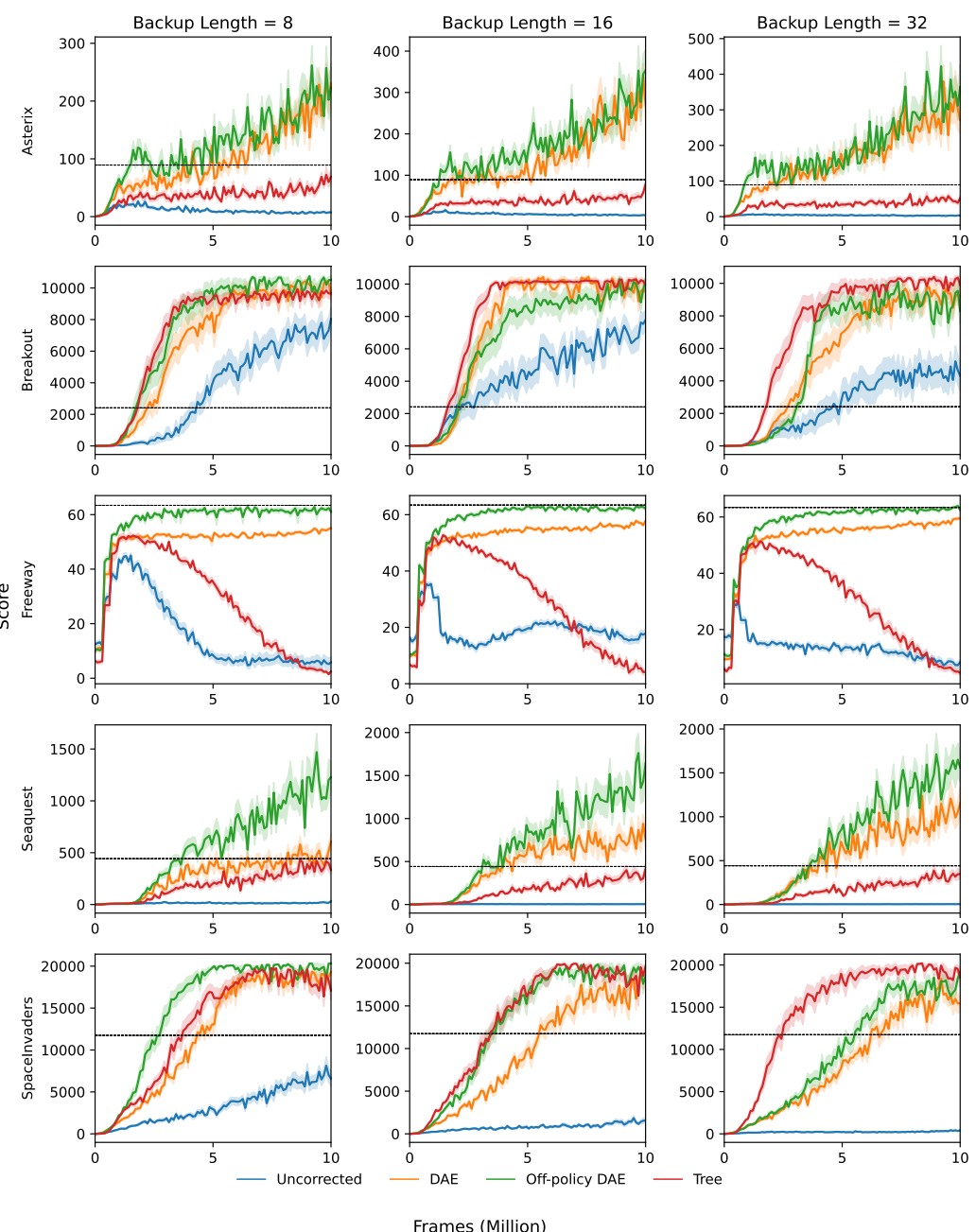

Figure 10: Training curves of each environment.

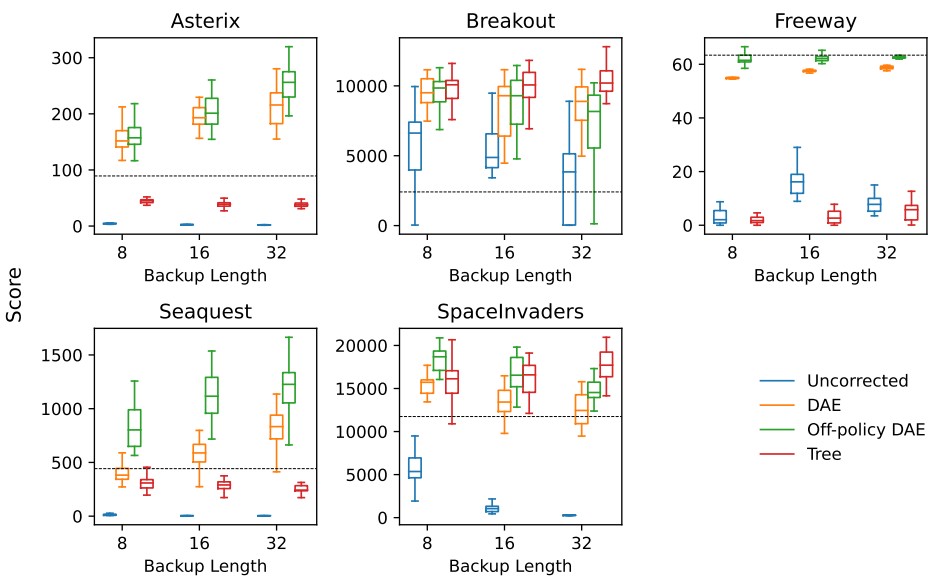

Figure 11: Distributions of the scores obtained by each method under different backup lengths.

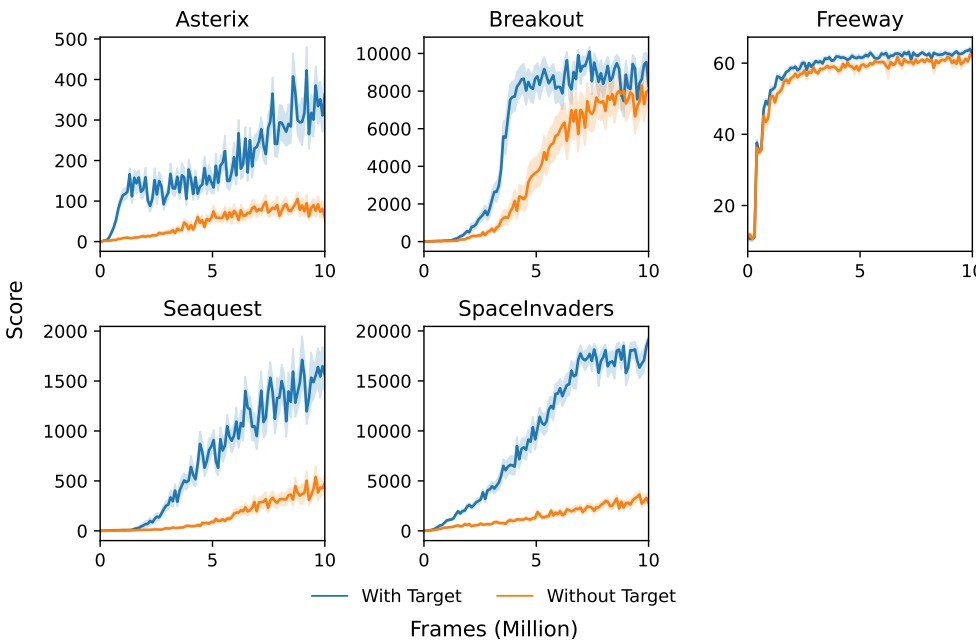

Figure 12: Off-policy DAE with and without target networks.

### D.7 COMPUTATIONAL RESOURCES

All experiments were performed on an internal cluster of NVIDIA A100 GPUs. Training an agent takes approximately 2 hours, depending on the backup method and the environment. For Off-policy DAE, the training time is significantly longer (approx. 15 hours) due to CVAE training. The increase in training time is largely due to the use of a large residual network for the CVAE, which we found to be easier to optimize compared to smaller convolutional networks.

