# OpenReview forum: "Skill or Luck? Return Decomposition via Advantage Functions"
_ICLR.cc/2024/Conference — ICLR 2024 poster_

### Official Review · Reviewer_K7qX · 2023-10-30

**Soundness:** 3 good
**Presentation:** 3 good
**Contribution:** 3 good
**Rating:** 6
**Confidence:** 3

**Summary:**

The proposes to decompose the advantage function into two parts: 1) return due to the agent's action selection and 2) return due to transition dynamics of the environment. They then use this decomposition to extend an existing algorithm DAE to the off-policy setting.

**Strengths:**

1. The proposed decomposition is straightforward, and I think is a useful idea to spark newer ideas.
2. The toy examples and Section 4 is useful.

**Weaknesses:**

See questions

**Questions:**

1. It is not clear to me why off-policy corrections are not necessary given that the sequence of rewards was generated by a different policy; it feels like frequency at which the sequences appear must be corrected for.
2. Related to above, it appears that the only "off-policyness" in Eqn 5 is the $\pi$-centered constraint. Why is this sufficient for the off-policy correction?
3. The decomposition seems related to exogenous and endogenous stochasticity. Is there a way to phrase the current work in that context? I'd also refer the authors to this paper that seems relevant: https://people.csail.mit.edu/hongzi/var-website/content/iclr-19.pdf
4. I am curious if once the advantage function is decomposed into skill and luck, is there a benefit to weighing each component differently? I would suspect that this leads to some bias in the policy ordering, but I am wonder if say the skill related component is too small, it may get overshadowed by the luck component, and the agent may not learn efficiently.
5. Related to above, I am curious how off-policy DAE performs as a function of environment stochasticity.
6. In Figure 3 and 4, it is unclear to me why all methods are able to produce similar mean estimates? Of course each is different in terms of their variance, but all are centered around the same mean which is a bit surprising.
7. What were the number of trials for the Figure 5 results? These should be mentioned.

---

> ### Author Response · Authors · 2023-11-16
>
> We thank the reviewer for the constructive feedback, please see our answers below.
>
> > It is not clear to me why off-policy corrections are not necessary given that the sequence of rewards was generated by a different policy; it feels like frequency at which the sequences appear must be corrected for.
>
> > Related to above, it appears that the only "off-policyness" in Eqn 5 is the
> -centered constraint. Why is this sufficient for the off-policy correction?
>
> The key observation is that the following equation holds for *any* trajectory
>
> $G = \sum_{t=0}^\infty \gamma^t (A^\pi(s_t, a_t) + \gamma B^\pi(s_t, a_t, s_{t+1}) + V^\pi(s_0)$.
>
> In other words, this decomposition is invariant to the sampling policy.
> In contrast, take MC methods as an example, they are based on *expectations* over trajectories, which is why additional corrections are required when the sampling distribution and the target distribution mismatch.
>
> One problem of the decomposition is that it is not unique for arbitrary $\hat{A}(s,a)$ and $\hat{B}(s,a,s’)$ (to see this, note that we can absorb $A^\pi$ into $\hat{B}$, such that $\hat{A}=0$ and $\hat{B}= B^\pi + A^\pi$ is also a solution). However, once we impose the centering constraints on $\hat{A}$ and $\hat{B}$, the decomposition becomes unique with solutions $A^\pi$ and $B^\pi$. For more details, please see appendix A for the proof.
>
> > The decomposition seems related to exogenous and endogenous stochasticity. Is there a way to phrase the current work in that context? I'd also refer the authors to this paper that seems relevant: https://people.csail.mit.edu/hongzi/var-website/content/iclr-19.pdf
>
> The exogenous stochasticity introduced by [1] is mostly concerned with a special class of MDPs (input-driven MDPs) where external inputs may also interact with state variables, causing additional stochasticity. If the external inputs are observable, then the environment reduces to an ordinary MDP. In this case, the external inputs can be seen as actions from nature, and nature's policy can be factorized into two components: the external input and the state transitions. For unobserved external inputs, the problem becomes partially observable (POMDP), which is beyond the scope of the present work.
>
> > I am curious if once the advantage function is decomposed into skill and luck, is there a benefit to weighing each component differently? I would suspect that this leads to some bias in the policy ordering, but I am wonder if say the skill related component is too small, it may get overshadowed by the luck component, and the agent may not learn efficiently.
>
> While we have demonstrated that the decomposition can improve value estimation, it is not clear whether it is optimal to optimize the policy based on the estimated advantage function. It is possible that by weighing components differently can lead to biased but lower variance targets for policy optimization.
>
> In the case where the luck component dominates, the problem is inherently difficult, as it means the agent’s actions have little influence on the returns, and the returns are very noisy. This makes it challenging to differentiate between good and bad actions, and additional care may be required.
>
> > Related to above, I am curious how off-policy DAE performs as a function of environment stochasticity.
>
> This can depend on several factors, such as whether the transition probabilities are known (see Figure 4 for comparison), or the backup lengths used. In the ideal case where the transition probabilities are known a priori along with infinite backup length, then we can expect similar convergence properties to a constrained linear regression as pointed out in Section 4.1.
>
> > In Figure 3 and 4, it is unclear to me why all methods are able to produce similar mean estimates? Of course each is different in terms of their variance, but all are centered around the same mean which is a bit surprising.
>
> We note that these experiments were performed in tabular, and fixed-policy settings. Therefore, all the estimators were proven to converge to the true values of the states in the limit.
>
> > What were the number of trials for the Figure 5 results? These should be mentioned.
>
> The results were aggregated over 20 random seeds. We have updated Figure 5 to include this information.
>
> [1] Mao, H., Venkatakrishnan, S. B., Schwarzkopf, M., & Alizadeh, M. (2018). Variance reduction for reinforcement learning in input-driven environments. arXiv preprint arXiv:1807.02264.

---

> > ### Comment · Reviewer_K7qX · 2023-11-16
> >
> > Thank you for the response. I am satisfied with the response. I will keep the score the same as it reflects an accurate evaluation. I do think this type of decomposition between impact from nature and agent is useful for future research.

---

### Official Review · Reviewer_NYwT · 2023-11-01

**Soundness:** 3 good
**Presentation:** 1 poor
**Contribution:** 3 good
**Rating:** 5
**Confidence:** 5

**Summary:**

This paper proposed a new return decomposition method via advantage functions, which contains the average return, the skill term, and the luck term.

**Strengths:**

The formulations of the paper are quite clear and easy to follow. Although I didn't get the main idea of the paper through the text, the formulations are quite clear for me to follow.





Generally, I think the paper contributes to giving some insight into the decomposition of the return.

**Weaknesses:**

Although I can get the main ideas from the formulations of the paper, the writing of the paper is poor.

- The introduction is not quite clear. Although it starts with an intuitive example, it doesn't contribute too much to get the idea of the paper.
- Section 3 lacks motivation, making it kind of hard to follow.



The experimental results are not very convincing.

- For Figure 9, Table 2, Figure 10, 11, 12, are the results in deterministic environments? Are there any results in stochastic environments?  I believe this is very important as the evaluation in stochastic environments is the main contribution of the paper.
- The authors used very large $N$ (8) for Uncorrected. However, in existing literature, it's known 3 or 5 are the best. Therefore, the comparison is unfair.
- What's the performance of 1-step DQN?
- The paper doesn't compare to other state-of-the-art multi-step off-policy method, such as Retrace($\lambda$) [1].



[1] Munos R, Stepleton T, Harutyunyan A, et al. Safe and efficient off-policy reinforcement learning[J]. Advances in neural information processing systems, 2016, 29.



The below claim is exaggerated. I didn't find a theorem that clearly proves the properties of faster convergence.

> We demonstrate that (Off-policy) DAE can be seen as generalizations of Monte-Carlo (MC) methods that utilize sample trajectories more efficiently to achieve faster convergence.





Minor comments:

- Near eq.7: The second term $E{[V^\pi(s_{t+1})|s_t,a_t]}$. What are the random variables of expectation? If it's $s_{t+1}$, please use another notation to differentiate it from the existing one using $s_{t+1}$.
- The complexity of the method seems very large (15 hours vs. 2 hours, as stated in D.7)

**Questions:**

- Why the decomposition of the return could benefit learning? Could the author give more insights into it?
- What's the probability of the sticky action for stochastic environments?

---

> ### Author Response · Authors · 2023-11-16
>
> We thank the reviewer for the constructive feedback, please see our answers below.
>
> > The introduction is not quite clear. Although it starts with an intuitive example, it doesn't contribute too much to get the idea of the paper.
>
> > Section 3 lacks motivation, making it kind of hard to follow.
>
> The lottery example in the introduction serves as an example to demonstrate that the return of a trajectory can be attributed to two sources: (1) actions from the agent (choosing a set of numbers), and (2) stochastic transitions (lottery drawing). Section 3 further develops this idea in a quantitative way by first analyzing deterministic environments and then generalizing to stochastic environments. We have made this connection more explicit at the beginning of Section 3.
>
> > For Figure 9, Table 2, Figure 10, 11, 12, are the results in deterministic environments? Are there any results in stochastic environments? I believe this is very important as the evaluation in stochastic environments is the main contribution of the paper.
>
> We note that the MinAtar suite includes both deterministic (Breakout, and Space Invaders) and stochastic (Asterix, Freeway, and Seaquest) environments. As such, the aforementioned figures and table already include results in both deterministic and stochastic environments. These were separately aggregated in Figure 5 for easier visualization.
>
> > The authors used very large N (8) for Uncorrected. However, in existing literature, it's known 3 or 5 are the best. Therefore, the comparison is unfair.
>
> We have included new results below for Uncorrected (N={3, 5}):
>
> | | Asterix | Breakout | Freeway | Seaquest | Space Invaders |
> |-| - | - | - |- |- |
> | Uncorr. (N=3) | $25.5 \pm 0.7$ |  $7157.7 \pm 465.1$ | $2.4 \pm 1.0$ |  $437.6 \pm 21.5$ | $8826.9 \pm 473.0$ |
> | Uncorr. (N=5) | $5.9 \pm 0.2$ |  $8123.1 \pm 610.1$ |  $0.0 \pm 0.0$  |  $34.0 \pm 1.4$ | $7353.5 \pm 436.2$ |
> | DAE (worst N) | $155.6 \pm 5.4$ | $8119.8 \pm 617.5$ | $55.1 \pm 0.1$ | $ 413.5 \pm 25.1$ | $12560.4 \pm 441.4$ |
> | OffDAE (worst N) | $161.5 \pm 5.9$ | $7372.1 \pm 582.0$ | $61.9 \pm 0.6$ | $839.4 \pm 48.3$ | $14970.3 \pm 348.9$ |
>
> (the worst N results are from Table 2)
>
> From the table, we see that while decreasing backup lengths may improve the performance of Uncorrected in some of the environments, it still largely underperforms Off-policy DAE. In addition, we would like to point out that it is not conclusive whether N = 3 or 5 is the best for Uncorrected. For example, in [1], the authors found that N = 10 or 20 can significantly outperform N = 5. In [2], it was also found that Uncorrected can achieve competitive performance for N up to 20.
>
> > What's the performance of 1-step DQN?
>
> As the aim of the experiments is to compare how different value estimators can impact policy optimization performance in a controlled setting, we have not included other deep RL algorithms as baselines, except for the on-policy PPO-DAE baseline, which serves as a validation that the proposed off-policy method is truly more sample efficient. Consequently, we believe the lack of 1-step DQN baseline does not invalidate our findings.
>
> > The paper doesn't compare to other state-of-the-art multi-step off-policy method, such as Retrace
>
> In the present work, we have only compared to Tree Backup as it is similar to Off-policy DAE in that, it also does not impose any constraint (aside from coverage of state-actions), or even need any knowledge about the behavior policy. In contrast, importance-sampling based methods (e.g., Retrace) require the behavior policy to compute the likelihood ratio, and often assumes the behavior policy to be Markovian, which we do not assume.
>
> > The below claim is exaggerated. I didn't find a theorem that clearly proves the properties of faster convergence.
>
> > *We demonstrate that (Off-policy) DAE can be seen as generalizations of Monte-Carlo (MC) methods that utilize sample trajectories more efficiently to achieve faster convergence.*
>
> We have removed “to achieve faster convergence” from the sentence. However, we would like to note that the toy experiments in section 4 were meant as empirical evidence that (Off-policy) DAE can achieve faster convergence.
>
> [1] Van Hasselt, H. P., Hessel, M., & Aslanides, J. (2019). When to use parametric models in reinforcement learning?. Advances in Neural Information Processing Systems, 32.
>
> [2] Hernandez-Garcia, J. F., & Sutton, R. S. (2019). Understanding multi-step deep reinforcement learning: A systematic study of the DQN target. arXiv preprint arXiv:1901.07510.

---

> ### Author Response · Authors · 2023-11-16
> **Official Comment by Authors (part 2)**
>
> > Near eq.7: The second term $E[V^\pi(s_{t+1})|s_t, a_t]$. What are the random variables of expectation? If it's $s_{t+1}$, please use another notation to differentiate it from the existing one using $s_{t+1}$
>
> We have updated the equations to explicitly point out the variable being integrated.
>
> > The complexity of the method seems very large (15 hours vs. 2 hours, as stated in D.7)
>
> We acknowledge that the computational cost of the proposed method can be significantly larger than other methods. In the present work, we have focused on the correctness of the method, and we will leave it for future work to explore more efficient ways to combine it with other deep RL algorithms.
>
> > Why the decomposition of the return could benefit learning? Could the author give more insights into it?
>
> Essentially, the decomposition enables better utilization of trajectories in comparison to MC methods, where only the starting state(-action)s are used. For example, in Figure 3 we can see that the rewards are caused by actions at state 3 for both starting states 1 and 2. However, MC methods ignore this relationship and estimate V(1) and V(2) independently. In contrast, DAE can account for the variance caused by the actions at state 3 through the advantage function, which is then shared between trajectories starting from either states 1 or 2. Off-policy DAE further extends this idea to include variance caused by stochastic transitions, as demonstrated in Figure 4.
>
> > What's the probability of the sticky action for stochastic environments?
>
> As the MinAtar suite already includes both deterministic and stochastic environments, we did not use sticky actions in the experiments.

---

> > ### Comment · Area_Chair_YPMx · 2023-11-20
> > **author reviewer discussion is ending soon**
> >
> > Dear NYwT,
> >
> > The author reviewer discussion period is ending soon this Wed. It looks the authors have provided both clarification and new empirical results. Does the author response clear your concerns or there are still outstanding questions that you would like the authors to address?
> >
> > Thanks again for your service to the community.
> >
> > Best,
> > AC

---

> ### Comment · Reviewer_NYwT · 2023-11-21
> **Response to the authors' response**
>
> Thanks for the response of the authors. And I'm happy to see the authors corrected some typos in the paper.
> Below are some further concerns.
>
> >We note that the MinAtar suite includes both deterministic (Breakout, and Space Invaders) and stochastic (Asterix, Freeway, and Seaquest) environments. As such, the aforementioned figures and table already include results in both deterministic and stochastic environments. These were separately aggregated in Figure 5 for easier visualization.
>
> Then, Figure 5 is misleading. The paper never clearly stated that the deterministic environments are referred to as "Breakout, and Space Invaders", and stochastic environments refer to as "Asterix, Freeway, and Seaquest".
> I thought they referred to using sticky action prob=0 and stick action prob=0.1 at the beginning.
>
> Minor Comments: In Breakout, the starting direction of the ball is also random. It's hard to say it's deterministic.
>
>
>
> > As the MinAtar suite already includes both deterministic and stochastic environments, we did not use sticky actions in the experiments.
>
> I believe that evaluating the methods under larger stochasticity is necessary because this is the major contribution of the method. This implies that using a non-zero sticky action prob is necessary for this paper. First, the original stochasticity is not enough. Second, the zero sticky action prob is usually not convincing to test the behavior of the algorithm. See below for the statements in the MinAtar paper.
>
> “This deterministic behavior can be exploited by simply repeating specific sequences of actions, rather than learning policies that generalize. Machado et al. (2017) address this by adding sticky-actions, where the environment repeats the last action with a probability 0.25 instead of executing the agent’s current action. We incorporate sticky-actions in MinAtar, but with
> a smaller probability of 0.1. ”
>
> Due to the efficiency and simplicity of MinAtar compared to classic Atari environments, I would expect the author not to reduce the difficulty of the benchmark and evaluate the methods with sufficient stochasticity.
>
> > We have included new results below for Uncorrected (N={3, 5}):
>
> As far as I know, Uncorrected DQN can perform quite well on Freeway and get a score of 60 even with N=3 (even under sticky action prob=0.1). I'm curious what happened to the 3-step Uncorred method here.  What's the result of 1-step method?

---

> > ### Author Response · Authors · 2023-11-21
> > **Response to Reviewer NYwT**
> >
> > We thank the reviewer for the additional feedback.
> >
> > > Then, Figure 5 is misleading. The paper never clearly stated that the deterministic environments are referred to as "Breakout, and Space Invaders", and stochastic environments refer to as "Asterix, Freeway, and Seaquest".
> >
> > Currently, we have included this information in the appendix under D.5, where we present the per-environment results.
> >
> > > Minor Comments: In Breakout, the starting direction of the ball is also random. It's hard to say it's deterministic.
> >
> > In this work, we mainly concern ourselves with stochastic *transitions*; therefore, we have classified Breakout as deterministic. We will update the appendix to reflect this.
> >
> > > I believe that evaluating the methods under larger stochasticity is necessary because this is the major contribution of the method...
> >
> > In the present work, we mainly follow the environment settings outlined in [1], which also do not use sticky actions. Additionally, we would like to point out that:
> >
> > (1) Even without sticky actions, we observe significant improvements by simply incorporating the luck terms (DAE -> Off-policy DAE, Fig 5), suggesting the positive impact of the luck terms in stochastic environments.
> >
> > (2) The stochasticity introduced by sticky actions violates the Markov property, as it depends on previous actions, rendering the environment partially observable (POMDP). Consequently, this aspect is beyond the scope of the present work.
> >
> > > As far as I know, Uncorrected DQN can perform quite well on Freeway and get a score of 60 even with N=3 (even under sticky action prob=0.1). I'm curious what happened to the 3-step Uncorred method here. What's the result of 1-step method?
> >
> > We believe this is partly due to previous works (e.g., [2, 3]) using much smaller replay buffers (0.1M frames) compared to ours (1M frames), which effectively makes the stored transitions more on-policy. We also observe similar effects from the per-environment training curves in Fig 10, where Uncorrected can initially perform on par with other methods (as transitions are mostly on-policy), but its performance deteriorates as more data is collected.
> >
> > Due to time constraints, we are unable to run new experiments for the case N=1. We kindly refer the reviewer to [3], where the authors performed a comparable analysis between N=1 and N=3 using DQN-based algorithms & MinAtar.
> >
> >
> > [1] Pan, H. R., Gürtler, N., Neitz, A., & Schölkopf, B. (2022). Direct advantage estimation. Advances in Neural Information Processing Systems, 35, 11869-11880.
> >
> > [2] Young, K., & Tian, T. (2019). Minatar: An atari-inspired testbed for thorough and reproducible reinforcement learning experiments. arXiv preprint arXiv:1903.03176.
> >
> > [3] Ceron, J. S. O., & Castro, P. S. (2021, July). Revisiting rainbow: Promoting more insightful and inclusive deep reinforcement learning research. In International Conference on Machine Learning (pp. 1373-1383). PMLR.

---

### Official Review · Reviewer_y5tB · 2023-11-03

**Soundness:** 4 excellent
**Presentation:** 4 excellent
**Contribution:** 3 good
**Rating:** 8
**Confidence:** 3

**Summary:**

This paper presents Off-policy DAE, a framework for off-policy learning that decomposes returns into two components: 1. those controllable by agents (skills) and 2. those beyond an agent's control (luck). Specifically, "luck" refers to stochastic transitions that the agent can't control. By explicitly modeling the advantage attributed to this luck factor, the agent can more effectively discern the impact of its own actions, leading to quicker generalization through enhanced credit assignment. Evaluations conducted on 5 MinAtar environments demonstrate improvements over baselines, particularly in scenarios with stochastic transitions.

**Strengths:**

S1: This offers a methodical approach to incorporate the effects of non-controllable factors, leading to enhanced credit assignment in off-policy learning. It's a novel concept, one I haven't encountered in other papers.

S2: The paper is very well-written and straightforward. I'm especially impressed by the intuitive examples provided to make the reader understand the main concept.

S3: Experiments are conducted on a standard benchmark suite using 20 random seeds. The results are notable, demonstrating that in stochastic environments, the agent performs better compared to methods not leveraging the proposed advantage function decomposition.

**Weaknesses:**

The paper is largely well-composed, with the proposed idea articulately presented. While I couldn't pinpoint any specific areas needing improvement,

W1: Conducting experiments in more domains might bolster the paper's claims even further.

**Questions:**

Q1: Can the method be directly applied to offline RL, and if so, what additional challenges might arise in that context?

---

> ### Author Response · Authors · 2023-11-16
>
> We thank the reviewer for the positive feedback, please see our answer to your question below.
>
> > Can the method be directly applied to offline RL, and if so, what additional challenges might arise in that context?
>
> One major challenge of offline RL lies in distribution shifts between target policies and behavior policies, causing agents to drift into less visited states in the dataset, where value estimates may be unreliable[1]. Previously, this was mitigated by using conservative/uncertainty estimation of the value functions [2, 3]. While in principle, the proposed method can also be applied to offline RL problems, we expect similar difficulties to arise. As such, an important question is how we can extend previous approaches to the proposed method.
>
> [1] Levine, S., Kumar, A., Tucker, G., & Fu, J. (2020). Offline reinforcement learning: Tutorial, review, and perspectives on open problems. arXiv preprint arXiv:2005.01643.
>
> [2] Kumar, A., Zhou, A., Tucker, G., & Levine, S. (2020). Conservative q-learning for offline reinforcement learning. Advances in Neural Information Processing Systems, 33, 1179-1191.
>
> [3] Ghasemipour, K., Gu, S. S., & Nachum, O. (2022). Why so pessimistic? estimating uncertainties for offline rl through ensembles, and why their independence matters. Advances in Neural Information Processing Systems, 35, 18267-18281.

---

> > ### Comment · Reviewer_y5tB · 2023-11-23
> >
> > Thank you for the response.

---

### Official Review · Reviewer_rop6 · 2023-11-06

**Soundness:** 3 good
**Presentation:** 4 excellent
**Contribution:** 3 good
**Rating:** 6
**Confidence:** 4

**Summary:**

This paper extends Direct Advantage Estimation (DAE), an algorithm designed to improve credit assignment by directly learning the advantage function. The original DAE formulation was limited to the on-policy case and this paper derives an off-policy version. This new algorithm is shown to be beneficial in a toy example and with experiments on the MinAtar environments.

**Strengths:**

- The decomposition of the return in terms of advantages of both the agent and the environment's "actions" is very intriguing and novel. It also leads to a practical algortihm, which could potentially outperform the standard approaches to learning advantage functions, a crucial part of actor-critic algorithms.

- The paper is well-written and this makes the derivation much easier to follow.
- The toy environments are well-designed to demonstrate the differences between the different approaches, DAE vs. off-policy DAE.
- There's some nice insights into why the uncorrected off-policy n-step return may work well in practice due to certain environmental properties.

**Weaknesses:**

- The larger-scale experiments are fairly limited with MinAtar being the most complex domain. Other environments could be considered.
- Other baselines could be more approrpirate for the MinAtar experiment. See questions.

**Questions:**

- Off-policy DAE requires an additional neural network to estimate the transition distribution. This seems slightly unelegant since actor-critic algorithms are usually model-free. Is it possible to avoid this by converting the constraint into a loss function instead? i.e. optimize the Lagrangian of the constraint with SGD? Perhaps we would have to use an approximate loss here to make it tractable.

- If we do learn a model to estimate $B_\pi$, is it possible to adapt the algorithm so the model is sampled-based only? Requiring a sum over all discrete latent states seems a bit restrictive in the choice of model and it seems to prevent easy scaling of the model size.

- Could you clarify how equation (10) reduces to the policy improvement lemma? In particular, what happens to the $B^\pi_t$ term?

- Why was Tree Backup chosen as the baseline? It seems like a less popular or effective choice compared to, say, Retrace [1] or its successors.

- It could be interesting to try to incorporate off-policy DAE with model-based RL methods since those algorithms would already have a learned model to use. E.g. Dreamer-v3

Minor point:
- To improve the clarity of the notation, I would suggest using capital letters for the random variables. In certain places, the lowercase letters are used to denote both fixed values and random ones. E.g. below equation (7), in the definition of $B^\pi_t$, $s_{t+1}$ in the expectation should be uppercase.


[1] "Safe and efficient off-policy reinforcement learning" Munos et al.

---

> ### Author Response · Authors · 2023-11-16
>
> We thank the reviewer for the constructive feedback, please see our answers below.
>
> > Off-policy DAE requires an additional neural network to estimate the transition distribution. This seems slightly unelegant since actor-critic algorithms are usually model-free. Is it possible to avoid this by converting the constraint into a loss function instead? i.e. optimize the Lagrangian of the constraint with SGD? Perhaps we would have to use an approximate loss here to make it tractable.
>
> This is indeed possible, and, in fact, conditional moment constraints is an active area of research in other domains such as econometrics and causal inference (e.g., [1], and [2] for its relationship to RL), where various methods have been developed to enforce such constraints. In the present work, however, we opt for the conceptually simpler approach by learning the transition probabilities, and leave it for future work to explore other more efficient possibilities.
>
> > If we do learn a model to estimate $B^\pi$, is it possible to adapt the algorithm so the model is sampled-based only? Requiring a sum over all discrete latent states seems a bit restrictive in the choice of model and it seems to prevent easy scaling of the model size.
>
> While we have not tried a sampling-based approach to estimate $B^\pi$, we do believe that approximating the sum (or integration) over the latent variables by Monte-Carlo methods is also a feasible approach. This may be especially useful in continuous control domains where the state transitions are better characterized by continuous distributions (e.g., robotics tasks).
>
> > Could you clarify how equation (10) reduces to the policy improvement lemma? In particular, what happens to the $B^\pi_t$ term?
>
> Firstly, we note that $B^\pi$ satisfies the centering property of the advantage function, i.e., $\mathbb{E}\_{s'}[B^\pi(s,a,s')|s,a]=\sum_{s’}B^\pi(s,a,s’)p(s'|s,a)=0$. This means that in expectation, all the $B^\pi_t$ terms vanish, leaving only the $G$, $A$, and $V$ terms, resulting in
> $V^\mu(s) = \mathbb{E}\_\mu[G|s_0=s] = \mathbb{E}\_\mu[\sum_{t=0}^\infty \gamma^t (A^\pi_t + \gamma B^\pi_t) + V^\pi(s_0)|s_0=s]
> = \mathbb{E}\_\mu[\sum_{t=0}^\infty \gamma^t A^\pi_t|s_0=s] + V^\pi(s)$.
>
> > Why was Tree Backup chosen as the baseline? It seems like a less popular or effective choice compared to, say, Retrace [1] or its successors.
>
> Tree Backup was chosen as the baseline as it is similar to Off-policy DAE in that, it also does not impose any constraint (aside from coverage of state-actions), or even need any knowledge about the behavior policy. In contrast, importance-sampling based methods (e.g., Retrace) require the behavior policy to compute the likelihood ratio, and often assumes the behavior policy to be Markovian, which we do not assume.
>
> > It could be interesting to try to incorporate off-policy DAE with model-based RL methods since those algorithms would already have a learned model to use. E.g. Dreamer-v3
>
> This is certainly an interesting direction, and we believe that existing model-based methods can benefit from the proposed approach to improve off-policy critic training. We will leave this for future work to explore the optimal way to combine Off-policy DAE and model-based methods.
>
> > To improve the clarity of the notation, I would suggest using capital letters for the random variables. In certain places, the lowercase letters are used to denote both fixed values and random ones. E.g. below equation (7), in the definition of in the expectation should be uppercase.
>
> We avoided using capital letters for the random variables because we were worried that it might lead to confusion between the advantage function and actions. As a remedy, we have updated the equations by explicitly pointing out the variable that is being integrated.
>
> [1] Kremer, H., Zhu, J. J., Muandet, K., & Schölkopf, B. (2022, June). Functional generalized empirical likelihood estimation for conditional moment restrictions. In International Conference on Machine Learning (pp. 11665-11682). PMLR.
>
> [2] Chen, Y., Xu, L., Gulcehre, C., Paine, T. L., Gretton, A., De Freitas, N., & Doucet, A. (2022). On instrumental variable regression for deep offline policy evaluation. The Journal of Machine Learning Research, 23(1), 13635-13674.

---

> > ### Comment · Reviewer_rop6 · 2023-11-22
> >
> > Thank you for the clarifications.
> > After reading the other reviews and responses, I still lean favorably towards this paper and will keep my current score.
> > In the future, I think investigating a model-free formulation of off-policy DAE could be impactful by simplifying the method, reducing its runtime and making it more scalable.

---

### Meta-Review · Area_Chair_YPMx · 2023-12-05

**Metareview:**

This paper proposes a novel decomposition of the return random variable into two parts. The first part is from the action selection of the agent and the second part is controlled by the stochasticity of the environment. Based on this decomposition, a novel objective is proposed to estimate the value function and the advantage of a policy in a multi-step off-policy manner. In practice, to minimize the objective, the paper proposes to use a variational approach to approximate the transition function. Performance improvements are seen in MinAtar environments.

Strength: The proposed return decomposition appears novel and sheds insights on the nature of the return random variable. The proposed objective opens up the possibility for future work. The performance improvement is clear in MinAtar.

Weakness: The proposed approach is essentially a model-based method since it requires to approximate the transition function. I thus feel it is necessary to compare the proposed method with model-based approaches, which are currently missing. MinAtar is still relatively small scale. It would be nice to investigate the efficacy of the proposed approach in larger-scale problems. Since MinAtar is already partially observable even without sticky actions, it also makes sense to add sticky actions as recommended by the authors of MinAtar to more thoroughly investigate the proposed approach. I, however, do note that I believe the novelty in this decomposition outweighs the defect in the empirical study.

**Justification For Why Not Higher Score:**

My major concern is that the comparison is not entirely fair since the proposed approach is essentially model-based while all baselines are model-free.

**Justification For Why Not Lower Score:**

The decomposition appears novel and could open up possibility for future works.

---

### Decision · Program_Chairs · 2024-01-16

Accept (poster)